# Mechanistic theory predicts the effects of temperature and humidity on inactivation of SARS-CoV-2 and other enveloped viruses

Dylan H Morris[1,2†]*, Kwe Claude Yinda[3†], Amandine Gamble[2†], Fernando W Rossine[1], Qishen Huang[4], Trenton Bushmaker[3], Robert J Fischer[3], M Jeremiah Matson[3,5], Neeltje Van Doremalen[3], Peter J Vikesland[4], Linsey C Marr[4], Vincent J Munster[3], James O Lloyd-Smith[2]*

[1]Department of Ecology and Evolutionary Biology, Princeton University, Princeton, United States; [2]Department of Ecology and Evolutionary Biology, University of California, Los Angeles, Los Angeles, United States; [3]Rocky Mountain Laboratories, National Institute of Allergy and Infectious Diseases, Hamilton, United States; [4]Department of Civil and Environmental Engineering, Virginia Tech, Blacksburg, United States; [5]Joan C. Edwards School of Medicine, Marshall University, Huntington, United States

**Abstract** Ambient temperature and humidity strongly affect inactivation rates of enveloped viruses, but a mechanistic, quantitative theory of these effects has been elusive. We measure the stability of SARS-CoV-2 on an inert surface at nine temperature and humidity conditions and develop a mechanistic model to explain and predict how temperature and humidity alter virus inactivation. We find SARS-CoV-2 survives longest at low temperatures and extreme relative humidities (RH); median estimated virus half-life is >24 hr at 10°C and 40% RH, but ~1.5 hr at 27°C and 65% RH. Our mechanistic model uses fundamental chemistry to explain why inactivation rate increases with increased temperature and shows a U-shaped dependence on RH. The model accurately predicts existing measurements of five different human coronaviruses, suggesting that shared mechanisms may affect stability for many viruses. The results indicate scenarios of high transmission risk, point to mitigation strategies, and advance the mechanistic study of virus transmission.

*For correspondence:
dylan@dylanhmorris.com (DHM);
jlloydsmith@ucla.edu (JOL-S)

†These authors contributed equally to this work

Competing interests: The authors declare that no competing interests exist.

## Introduction

For viruses to transmit from one host to the next, virus particles must remain infectious in the period between release from the transmitting host and uptake by the recipient host. Virus environmental stability thus determines the potential for surface (fomite) transmission and for mid-to-long range transmission through the air. Empirical evidence suggests that virus environmental stability depends strongly on ambient temperature and humidity, particularly for enveloped viruses; examples among enveloped viruses that infect humans include influenza viruses (*Marr et al., 2019*), endemic human coronaviruses (*Ijaz et al., 1985*), and the zoonotic coronaviruses SARS-CoV-1 (*Chan et al., 2011*) and MERS-CoV (*van Doremalen et al., 2013*).

In late 2019, a new zoonotic coronavirus now called SARS-CoV-2 emerged; it has since caused a global pandemic (COVID-19) and is poised to become an endemic human pathogen. Many countries in the Northern Hemisphere experienced a substantial uptick in transmission with the arrival of their late autumn and winter. Epidemiologists had anticipated such a seasonal increase (*Neher et al.,*

*2020*; *Kissler et al., 2020*) based on observations from other enveloped respiratory viruses, such as endemic human coronaviruses (*Monto et al., 2020*) and influenza viruses (*Lofgren et al., 2007*). These viruses spread more readily in temperate zone winters than in temperate zone summers. SARS-CoV-2 has also displayed epidemic dynamics shaped by superspreading events, in which one person transmits to many others (*Furuse et al., 2020*; *Kain et al., 2021*); the related SARS-CoV-1 virus was likewise characterized by superspreading (*Lloyd-Smith et al., 2005*).

Virus transmission is governed by many factors, among them properties of the virus and properties of the host population. But anticipating seasonal changes in transmission and preventing superspreading events both require an understanding of virus persistence in the environment, since ambient conditions can facilitate or impede virus spread. Empirical evidence suggests that SARS-CoV-2, like other enveloped viruses, varies in its environmental stability as a function of temperature and humidity (*Biryukov et al., 2020*; *Matson et al., 2020*), but the joint effect of these two factors remains unclear.

Moreover, despite years of research on virus environmental stability, there do not exist mechanistically motivated quantitative models for virus inactivation as a function of both temperature and humidity. Existing predictive models for the environmental stability of SARS-CoV-2 (*Biryukov et al., 2020*; *Guillier et al., 2020*) and other viruses (*Posada et al., 2010*) are phenomenological regression models that do not model the underlying biochemical mechanisms of inactivation. This limits both our insight into the underlying inactivation process and our ability to generalize from any given experiment to unobserved conditions, or to real-world settings. A lack of quantitative, mechanistic models also makes it difficult to determine which environmental factors are most important, for instance whether absolute humidity (*Shaman et al., 2010*) or relative humidity (*Marr et al., 2019*) best explains influenza inactivation and seasonality.

We measured the environmental stability of SARS-CoV-2 virus particles (virions) suspended in cell culture medium and deposited onto a polypropylene plastic surface at nine environmental conditions: three relative humidities (RH; 40%, 65%, and 85%) at each of three temperatures (10°C, 22°C, and 27°C). We first quantified viable (infectious) virus titer over time and estimated virus decay rates and corresponding half-lives in each condition using a simple Bayesian regression model (see Materials and methods). We quantified the evaporation of the suspension medium and compared virus stability during the sample evaporation phase—while substantial water loss was ongoing—to virus stability after a quasi-equilibrium phase was reached—when further evaporation was not evident over the timescale of the experiment.

We then created a mechanistic biochemical model of virus inactivation kinetics, drawing upon existing hypotheses for how temperature and humidity affect the inactivation chemistry of virus particles in microdroplets (*Marr et al., 2019*; *Lin and Marr, 2020*). We fit this mechanistic model to our SARS-CoV-2 data, and used it to predict observations from other human coronaviruses and other studies of SARS-CoV-2, and to extrapolate our SARS-CoV-2 results to unobserved temperature and humidity conditions.

Our mechanistic model is based on a simple premise: virus inactivation in the environment is a chemical reaction and so obeys the laws of chemical kinetics. Reactions proceed faster at higher temperatures and higher solute concentrations. Solutes will be more concentrated when there is more evaporation; this occurs when the ambient relative humidity is lower. But below a threshold relative humidity, the efflorescence relative humidity (ERH), droplets may crystallize; this is also expected to change reaction kinetics. These principles apply across reactions. We do not need to know the exact identities and concentrations of non-virus reactants (e.g. amino acids, electrolytes, etc.) involved to make mechanistic predictions about how the inactivation reaction rate will vary with temperature and humidity.

Our model encodes these principles. We estimated its three central parameters from our data.

## Results

### Empirical patterns of virus decay

Our data suggest that SARS-CoV-2 environmental persistence could vary meaningfully across the range of temperatures and humidities encountered in daily life, with posterior median [95% credible interval] half-lives as long as 27 hr [20, 39] (10°C, 40% RH) and as short as 1.5 hr [1.1, 2.1] (27°C, 65%

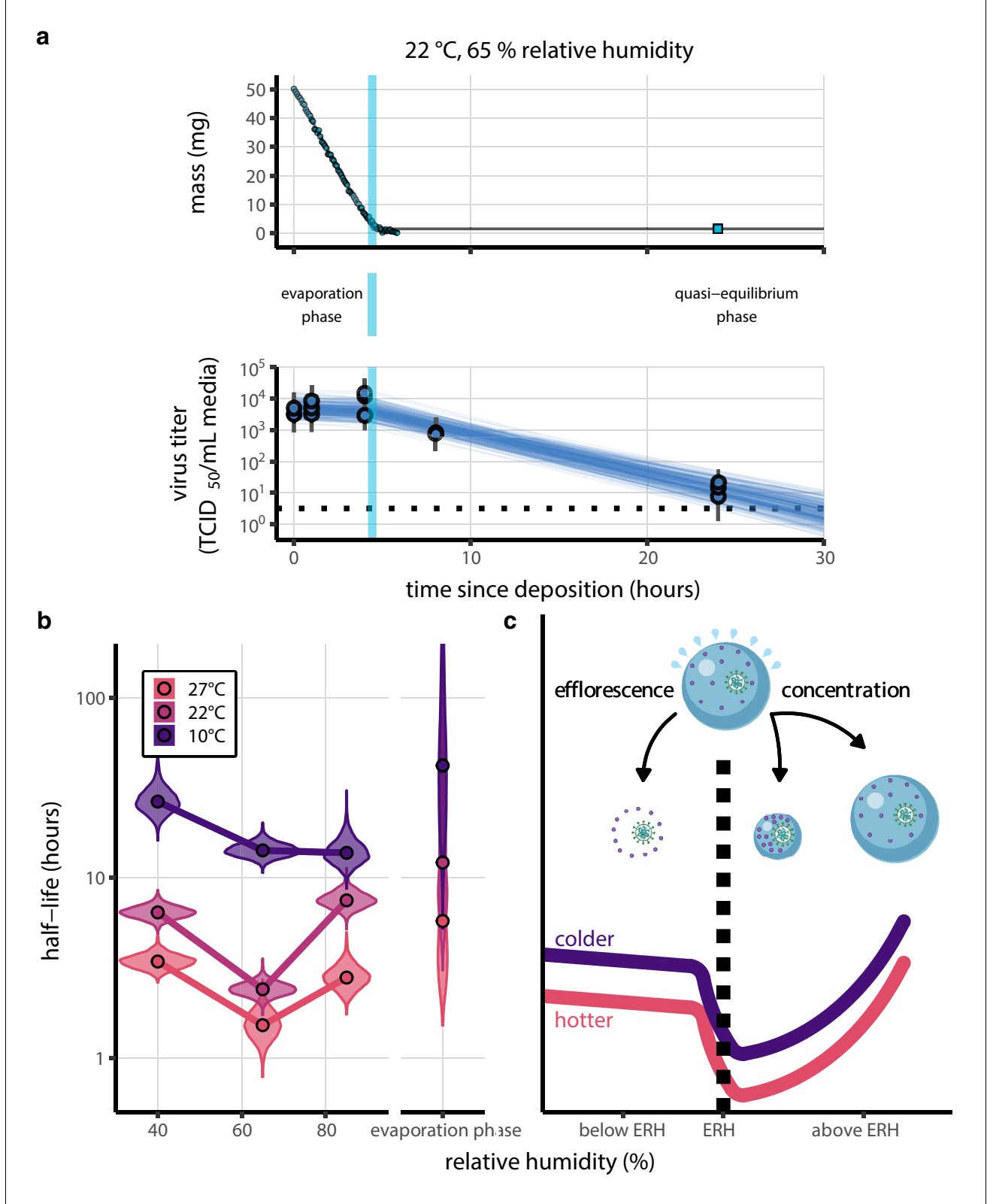

**Figure 1.** Inactivation kinetics and estimated half-life of SARS-CoV-2 on an inert surface as a function of temperature and relative humidity (RH). (a) Example of medium evaporation and virus inactivation as a function of time since deposition; experiments at 22°C and 65% RH shown. Inactivation proceeds in two phases: an evaporation phase during which water mass is lost from the sample to evaporation and a quasi-equilibrium phase once the sample mass has plateaued. Light blue vertical line shows posterior median estimated time that quasi-equilibrium was reached. Top plot: medium

*Figure 1 continued on next page*

*Figure 1 continued*

evaporation. Dots show measured masses. Square shows measured final (quasi-equilibrium) mass; plotted at 24 hr for readability. Lines are 10 random draws from the posterior for the evaporation rate; horizontal section of line reflects the reaching of quasi-equilibrium (measured final mass). See figure supplements for all conditions. Bottom plot: virus inactivation. Points show posterior median estimated titers in $\log_{10}$ TCID$_{50}$/mL for each sample; lines show 95% credible intervals. Black dotted line shows the approximate single-replicate limit of detection (LOD) of the assay: $10^{0.5}$TCID$_{50}$/mL media. Three samples collected at each time-point. Lines are 10 random draws per measurement from the posterior distribution for the inactivation rates, estimated by a simple regression model (see Materials and methods). (b) Measured virus half-lives. Violin plots show posterior distribution of estimated half-lives, plotted on a logarithmic scale. Dots show posterior median value. Color indicates temperature. Measurements at 40%, 65%, and 85% RH reflect decay kinetics once the deposited solution has reached quasi-equilibrium with the ambient air. Estimated half-lives for the evaporation phase that occurs prior to quasi-equilibrium are plotted to the right, since conditions during this phase are mainly dilute, and thus analogous to high RH quasi-equilibrium conditions. See figure supplements for plots showing the fit of the regression used to estimate half-lives to the titer data. (c) Schematic of hypothesized effects of temperature and relative humidity on duration of virus viability. Virus half-lives are longer at lower temperatures, regardless of humidity, because inactivation reaction kinetics proceed more slowly. Relative humidity affects virus half-life by determining quasi-equilibrium solute concentration in the droplet containing the virus. Above the efflorescence relative humidity (ERH), solutes are concentrated by evaporation. The lower the ambient humidity, the more water evaporates, the more concentration occurs, and the faster inactivation reactions proceed. Below the ERH, solutes effloresce, forming crystals. Half-lives are thus not particularly sensitive to changes in sub-ERH relative humidity, and half-lives even slightly below the ERH may be substantially longer than half-lives slightly above it.

The online version of this article includes the following figure supplement(s) for figure 1:

**Figure supplement 1.** Fit of the regression model used to estimate the half-lives in **b** to evaporation phase (pre-drying) SARS-CoV-2 titer data.
**Figure supplement 2.** Fit of the regression model used to estimate the half-lives in **b** to quasi-equilibrium (post-drying) SARS-CoV-2 titer data.
**Figure supplement 3.** Results of medium evaporation experiments.

RH), once droplets reach quasi-equilibrium with the ambient air conditions (*Figure 1b*, *Appendix 1— table 1*).

Minimal virus decay occurred during the evaporation phase (*Figure 1a*, *Figure 1—figure supplement 1*), when excess water was present. Estimated half-lives were long but exact values were highly uncertain, as the small amount of absolute virus inactivation during the brief evaporation phases, combined with the noise involved in sampling and titration, limits our inferential capacity. Posterior median half-lives during the evaporation phase were 42 hr [11, 330] at 10°C, 12 hr [4.5, 160] at 22°C, and 5.8 hr [2.1, 130] at 27°C (*Table 1*).

Overall, virus decay became markedly faster as temperature increased for all humidities, with decay at 27°C roughly five to ten times faster than decay at 10°C. Across temperatures, virus decay was relatively rapid at 65% RH and tended to be slower either at lower (40%) or higher (85%) humidities, or when excess water was present during the evaporation phase (*Figure 1b*, *Table 1*).

**Table 1.** Estimated half-lives in hours of SARS-CoV-2 on polypropylene as a function of temperature (T) and relative humidity (RH).
Estimated half-lives are reported as posterior median and the middle 95% credible interval.

|  | T (°C) | RH (%) | Median half-life (h) | 2.5 % | 97.5 % |
|---|---|---|---|---|---|
| Quasi-equilibrium phase | 10 | 40 | 26.55 | 20.28 | 38.75 |
|  | 10 | 65 | 14.22 | 12.17 | 17.16 |
|  | 10 | 85 | 13.78 | 10.67 | 19.70 |
|  | 22 | 40 | 6.43 | 5.52 | 7.56 |
|  | 22 | 65 | 2.41 | 2.03 | 2.88 |
|  | 22 | 85 | 7.50 | 6.22 | 9.24 |
|  | 27 | 40 | 3.43 | 2.91 | 4.12 |
|  | 27 | 65 | 1.52 | 1.05 | 2.14 |
|  | 27 | 85 | 2.79 | 2.12 | 3.78 |
| Evaporation phase | 10 |  | 42.08 | 10.97 | 334.34 |
|  | 22 |  | 12.18 | 4.47 | 163.58 |
|  | 27 |  | 5.76 | 2.14 | 125.85 |

## Mechanistic model for temperature and humidity effects

Many viruses, including SARS-CoV-2, exhibit exponential decay on surfaces and in aerosols (*Marr et al., 2019*; *van Doremalen et al., 2020*; *Biryukov et al., 2020*). We drew upon chemical principles of droplet evaporation and virus inactivation (*Figure 1c*) to create a minimal mechanistic model incorporating the effects of both temperature and relative humidity on exponential decay rates.

We model virus inactivation at quasi-equilibrium on inert surfaces as a chemical reaction with first-order reaction kinetics; that is, the quantity of virus is the limiting reactant of the rate-determining step. This reflects the empirical pattern of exponential decay and is consistent with the fact that virions will be numerically rare in microdroplets compared to other reactants.

We characterize the temperature dependence of this reaction with the Arrhenius equation, which describes a reaction rate (here the virus inactivation rate $k$) as a function of an activation energy $E_a$, an asymptotic high-temperature reaction rate $A$, the universal gas constant $R$, and the absolute temperature $T$:

$$k = A \exp\left(-\frac{E_a}{RT}\right) \tag{1}$$

Prior work has found Arrhenius-like temperature dependence for virus inactivation on surfaces and in aerosols for many viruses (*Adams, 1949*), including human coronaviruses (*Yap et al., 2020*).

Mechanistic principles of virus inactivation as a function of humidity have been more elusive. Recent work has suggested that relative humidity affects virus inactivation by controlling evaporation and thus governing the solute concentrations in a droplet containing virions (*Marr et al., 2019*; *Lin and Marr, 2020*). In more humid environments, evaporation is slower and more water remains when quasi-equilibrium is reached. In less humid environments, evaporation is faster and little or no water remains (*Figure 1c*).

When released from infected hosts, virions are found in host bodily fluids, and virus inactivation experiments are typically conducted in cell culture medium. Both solutions contain amino acids and electrolytes, in particular sodium chloride (NaCl) (*Cavaliere et al., 1989*; *Dulbecco and Freeman, 1959*). Prior work has found that higher quasi-equilibrium solute concentrations are associated with faster virus inactivation rates (*Yang and Marr, 2012*; *Yang et al., 2012*). The simplest explanation for this is that the measured solute concentration is a direct proxy for the concentration of the reactants governing the inactivation reaction. Thus, ambient humidity affects the reaction rate by setting the quasi-equilibrium concentrations of the reactants that induce inactivation of the virus.

The exact quasi-equilibrium state reached will depend on the solutes present, since different solutes depress vapor pressure to different degrees. In electrolyte solutions like bodily fluids or cell culture media, efflorescence is also important. Below a threshold ambient humidity—the efflorescence relative humidity (ERH)—electrolytes effloresce out of solution, forming a crystal (*Figure 1c*). Below the ERH, the reaction no longer occurs in solution, and so inactivation may be slower. The non-monotonic ('U-shaped') dependence of virus inactivation on relative humidity, observed in our data (*Figure 1a*) and elsewhere in the literature (*Yang et al., 2012*; *Benbough, 1971*; *Prussin et al., 2018*; *Webb et al., 1963*), including for coronaviruses (*Casanova et al., 2010*; *Songer, 1967*), could be explained by this regime shift around the ERH (*Figure 1c*).

During the evaporation phase prior to quasi-equilibrium, reactants are less concentrated and decay is expected to be slower, as observed from our data (*Figure 1a,b*). If small initial droplet sizes are used—as in real-world depositions (predominantly < 10 µL; *Johnson et al., 2011*; *Johnson et al., 2013*; *Thompson et al., 2013*) and in some experiments—evaporative quasi-equilibration should be near instant, and so inactivation should follow the kinetics at quasi-equilibrium. Larger droplets, such as those used in our experiments, will take more time to equilibrate (depending on temperature and humidity); this allows us to distinguish the quasi-equilibrium phase from the evaporation phase.

We partition inactivation at quasi-equilibrium into two humidity regimes, effloresced and solution, according to whether the ambient RH is below the ERH (effloresced) or above (solution). In either case, we approximate virus inactivation as a first-order reaction with inactivation rate $k_{\mathrm{eff}}$ or $k_{\mathrm{sol}}$, respectively. Based on observations of NaCl solutions at room temperature and atmospheric

pressure (*Mikhailov et al., 2004*), we use an ERH of 45%. This means that 40% RH experiments are in the effloresced regime and 65% and 85% RH experiments are in the solution regime.

We model the effloresced and solution inactivation rates $k_{\mathrm{eff}}$ and $k_{\mathrm{sol}}$ using two Arrhenius equations with a shared activation energy $E_a$ but distinct asymptotic high-temperature reaction rates $A_{\mathrm{eff}}$ and $A_{\mathrm{sol}}$. In solution conditions, we further modulate $k_{\mathrm{sol}}$ by a quasi-equilibrium 'concentration factor' $\frac{[S_{\mathrm{eq}}]}{[S_0]}$, which quantifies how concentrated the solution has become at quasi-equilibrium $[S_{\mathrm{eq}}]$ relative to its initial state $[S_0]$.

Given our assumption of first-order kinetics, an n-fold increase in the non-virion reactant concentrations should translate directly into an n-fold increase in the inactivation rate. Lower relative humidity leads to higher quasi-equilibrium concentration and thus increases virus inactivation rate, until the ERH is reached. Below the ERH, inactivation rates may again be low due to crystallization, depending on $A_{\mathrm{eff}}$. We do not force the relationship between RH and inactivation rate to be continuous at the ERH; there may be a discontinuity (see Appendix, Interpretation of the transition in inactivation rate at the ERH, for a discussion).

$$k_{\mathrm{eff}} = A_{\mathrm{eff}} \exp\left(-\frac{E_a}{RT}\right) \tag{2}$$

$$k_{\mathrm{sol}} = \frac{[S_{\mathrm{eq}}]}{[S_0]} A_{\mathrm{sol}} \exp\left(-\frac{E_a}{RT}\right) \tag{3}$$

We estimated $E_a$, $A_{\mathrm{eff}}$, and $A_{\mathrm{sol}}$ from our data, constraining all to be positive. We treated evaporation phase data as governed by $k_{\mathrm{sol}}$, with a dynamic value of the concentration factor $\frac{[S(t)]}{[S_0]}$ (Appendix, Modeling of virus decay dynamics during the evaporation phase). We computed the quasi-equilibrium concentration factor $\frac{[S_{\mathrm{eq}}]}{[S_0]}$ by fitting a theoretically-motivated curve to our evaporation data (*Figure 2—figure supplement 1*).

The relationship between RH and quasi-equilibrium concentration factor depends on complex evaporative kinetics that will vary among media. For this reason, we do not attempt to predict it from first principles, but instead measure it directly and use the fitted curve to extrapolate to unmeasured RH conditions. We use this approach for the results presented in the main text; we refer to it as the 'main model'.

To check robustness of the main model results, we also estimated a version of the model without this theoretical curve–using only directly-measured equilibrium concentration factors. This model (referred to as the 'directly-measured concentration model') yielded similar results to the main model; see Appendix, Mechanistic model versions for details.

We also considered a four-parameter variant of the model with distinct activation energies below the ERH ($E_a^{\mathrm{eff}}$) and above ($E_a^{\mathrm{sol}}$), placing the same prior on each. This accounts for the possibility that the rate-determining step of the inactivation reaction might be distinct in the two regimes. The estimated activation energies were very similar below and above the ERH (*Appendix 1—figure 1*). This suggests that the rate-determining reaction step—and thus the activation energy—is the same in both regimes. Accordingly, we report estimates from the three-parameter model with a shared $E_a$. We provide additional details and interpretation of our mechanistic inactivation modeling in the Appendix; see Mechanistic inactivation model interpretation and Mechanistic model estimation.

## Model fitting and prediction of unobserved conditions

Our dataset comprises nine experimental conditions, each with seven time-points that span the evaporation and quasi-equilibrium phases. We sought to explain the virus inactivation rates across this entire dataset using our mechanistic model with just three free parameters: the activation energy $E_a$ and the asymptotic high-temperature reaction rates under effloresced and solution conditions, $A_{\mathrm{eff}}$ and $A_{\mathrm{sol}}$. The mechanistic function used and the constraint on the parameters to be positive means that inactivation rate must increase with temperature and with increasing solute concentration. Remarkably, the fit of the mechanistic model (*Figure 2*) is nearly as good as that of the simple regression, in which we estimate independent exponential decay rates for each condition to measure virus half-life (*Figure 1—figure supplement 2*, see Appendix, Simple regression model). Mechanistic model parameter estimates are given in *Figure 2—figure supplement 2* and *Appendix 1—table 1*.

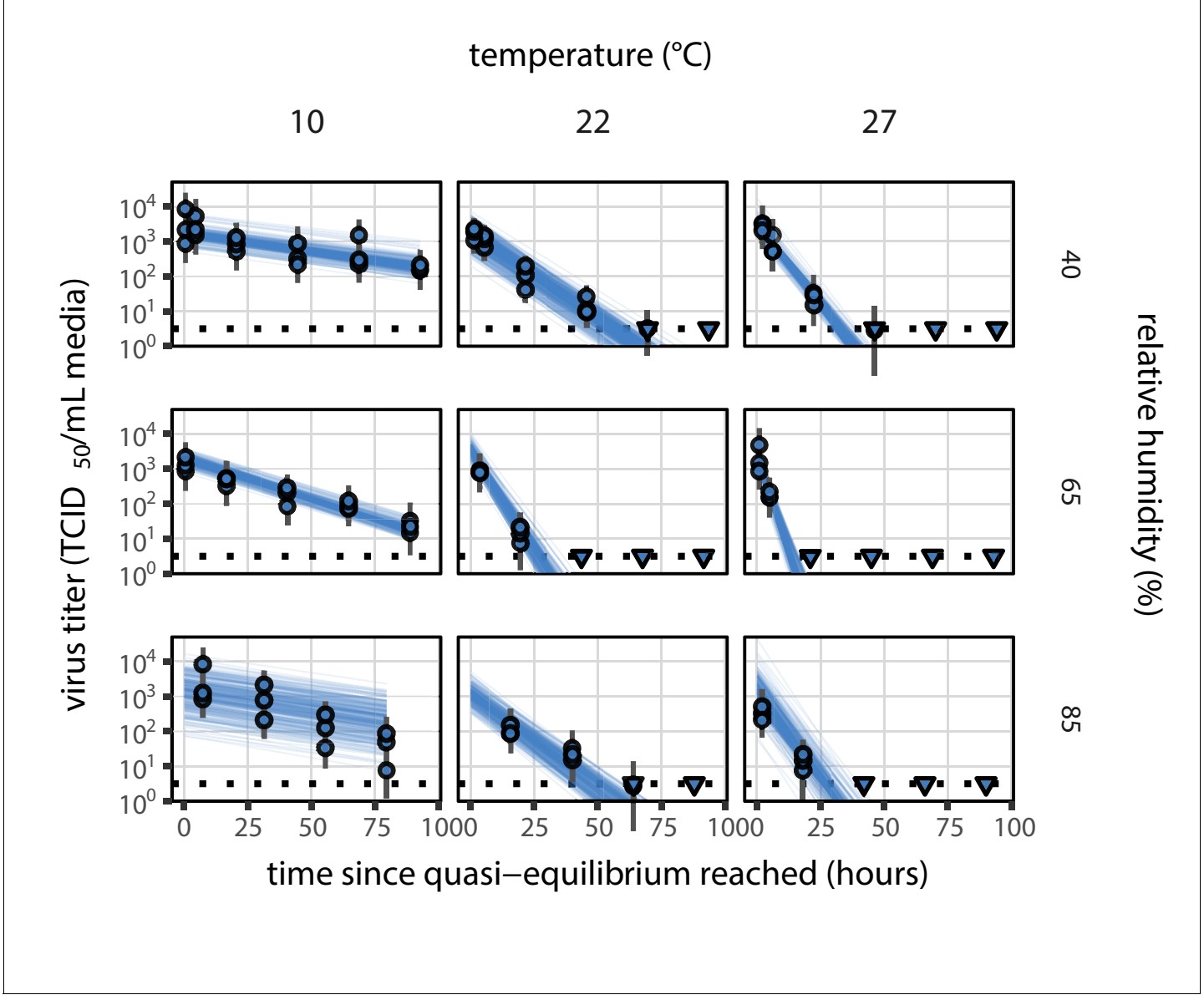

**Figure 2.** Estimated titers and main mechanistic model fit for SARS-CoV-2 stability on polypropylene at quasi-equilibrium. Points show posterior median estimated titers in $\log_{10}$ $TCID_{50}$/mL for each sample; lines show 95% credible intervals. Time-points with no positive wells for any replicate are plotted as triangles at the approximate single-replicate limit of detection (LOD) of the assay—denoted by a black dotted line at $10^{0.5}TCID_{50}$/mL media—to indicate that a range of sub-LOD values are plausible. Three samples collected at each time-point. x-axis shows time since quasi-equilibrium was reached, as measured in evaporation experiments. Lines are random draws (10 per sample) from the joint posterior distribution of the initial sample virus concentration and the mechanistic model predicted decay rate; the distribution of lines gives an estimate of the uncertainty in the decay rate and the variability of the initial titer for each experiment. See *Figure 2—figure supplement 4* for a visualization of the mechanistic model fit using directly-measured concentration, rather with a curve estimating the humidity/concentration relationship.

The online version of this article includes the following figure supplement(s) for figure 2:

**Figure supplement 1.** Fitted curve estimating the relationship between humidity and quasi-equilibrium concentration factor.

**Figure supplement 2.** Mechanistic model parameter estimates, both with and without a fitted curve relating RH to concentration.

**Figure supplement 3.** Comparison of directly-measured half-lives with those predicted by the mechanistic model, both with and without a fitted curve relating RH to concentration.

**Figure supplement 4.** Equivalent quasi-equilibrium phase figure, but using directly-measured concentration factors rather than a fitted curve that relates RH to concentration.

**Figure supplement 5.** Equivalent main mechanistic model fit figure for the evaporation phase.

**Figure supplement 6.** Evaporation phase figure, but using directly-measured concentration factors rather than a fitted curve that relates RH to concentration.

We used the mechanistic model to predict SARS-CoV-2 half-life for unobserved temperature and humidity conditions from 0°C to 40°C, and from 0% to 100% RH. We chose these ranges to reflect environments encountered by human beings in daily life. We did not extrapolate to temperatures below 0°C since inactivation kinetics may be different when fluid containing the virus freezes. The exact freezing points of suspension medium and human fluids at sea level will depend on solute concentration, but will typically be below the 0°C freezing point of pure water.

Median predicted SARS-CoV-2 half-life varies by more than three orders of magnitude, from less than half an hour at 40°C just above the modeled approximate ERH, to more than a month at 0°C and 100% RH (*Figure 3a,c*). We find good qualitative agreement between model predictions and model-free estimates from our data, including long half-lives prior to quasi-equilibrium. The U-shaped effect of humidity on virus half-life is readily explained by the regime-shift at the ERH (*Figure 3a*). In particular, half-lives become extremely long at cold temperatures and in very dilute solutions, which are expected at high RH (*Figure 3a,b*). Of note, the worst agreement between mechanistic model predictions and (independent) simple regression estimates is found at 10°C and 85% RH (*Figure 3a*). This is partially explained by the fact that the empirical quasi-equilibrium concentration reached under those conditions was higher than our model prediction based on RH (*Figure 2—figure supplement 1*). Accordingly, the half-life prediction for 10°C and 85% RH based on directly-measured concentrations is superior to the prediction based on an extrapolation from the relative humidity (*Figure 2—figure supplement 3*).

As a stronger test of our model's validity, we used our estimated $E_a$ and $A$ values to make out-of-sample predictions of the half-lives of five human coronaviruses reported from independent studies: four betacoronaviruses (SARS-CoV-2, SARS-CoV-1, MERS-CoV and HCoV-OC43) and one alphacoronavirus (HCoV-229E). We compiled data on the environmental stability of those viruses under conditions ranging from 4°C to 95°C, from 30% to 80% RH, and on a range of surfaces or bulk media, and computed empirical (regression) estimates of virus half-lives (*Appendix 1—table 3*). We also included data on stability of SARS-CoV-1 (*van Doremalen et al., 2020*) and MERS-CoV (same method as in *van Doremalen et al., 2020*) collected by our group during previous studies (*Appendix 1—table 4*).

Where both temperature and RH were available, we compared these model-free estimates to predictions based on the mechanistic model parameterized with our SARS-CoV-2 data (*Figure 3c*, *Figure 3—figure supplement 1*). We found striking agreement for half-life estimates both above and below the ERH, and for temperatures ranging from 4°C to 37°C.

To include a broader range of conditions in our out-of-sample model testing, we used our model to predict half-lives observed in all comparable studies by extrapolating from a reference half-life in each study. Predicted half-lives matched observations well across five orders of magnitude (*Figure 3d*), despite spanning five virus species and despite important heterogeneities in the data collection process (see Appendix, Meta-analysis of human coronavirus half-lives). The two conspicuous outliers, where SARS-CoV-2 half-lives were measured to be substantially shorter than our prediction, correspond to samples exposed to high heat in closed vials (*Chin et al., 2020*; Chin, 2020, personal communication) which is known to accelerate virus inactivation (*Gamble et al., 2021*).

## Discussion

Combining novel data, mathematical modeling, and a meta-analysis of existing literature, we have developed a unified, mechanistic framework to quantify the joint effects of temperature and humidity on virus stability. In particular, our model provides a mechanism for the non-linear and non-monotonic relationship between relative humidity and virus stability previously observed for numerous enveloped viruses (*Yang and Marr, 2012*; *Casanova et al., 2010*; *Songer, 1967*), but not previously reported for SARS-CoV-2. Our work documents and explains the strong dependence of SARS-CoV-2 stability on environmental temperature and relative humidity, and accurately predicts half-lives for five coronavirus species in conditions from 4°C to 95°C, and from 30% to 80% RH and in bulk solution.

Our findings have direct implications for the epidemiology and control of SARS-CoV-2 and other enveloped viruses. The majority of SARS-CoV-2 clusters have been linked to indoor settings (*Leclerc et al., 2020*), suggesting that virus stability in indoor environmental conditions may be an

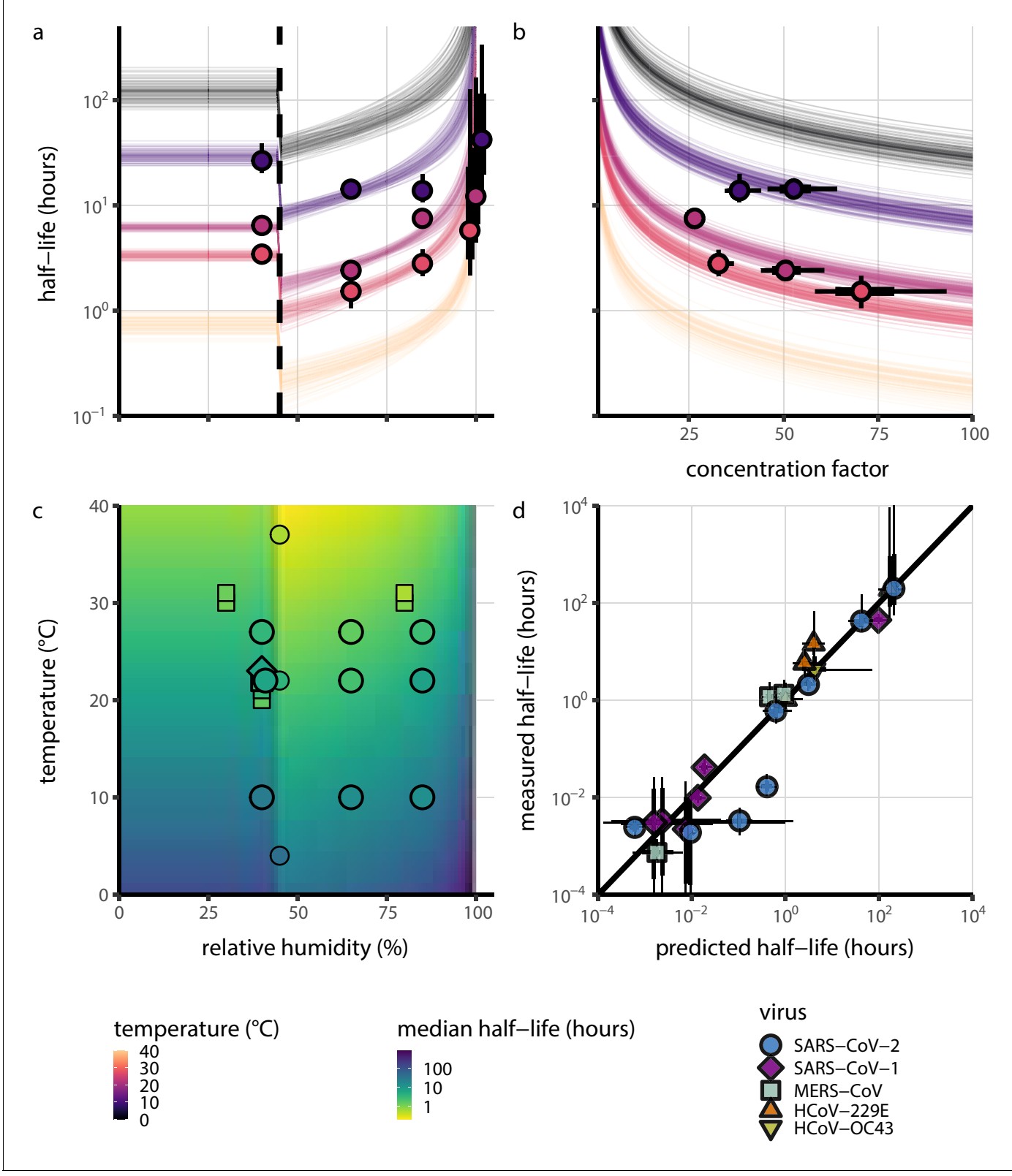

**Figure 3.** Extrapolation of human coronavirus half-life from the mechanistic model to unobserved temperatures and humidities and prediction of data from the literature. (a) Predicted half-life as a function of relative humidity. Points show posterior median for measured half-lives, estimated without the mechanistic model (simple regression estimate for each temperature/humidity combination), lines show a 68% (thick) and 95% (thin) credible interval. Dashed line shows the ERH. Estimated evaporation phase half-lives are plotted at the right. Colored lines show predicted half-lives as a function of

*Figure 3 continued on next page*

*Figure 3 continued*

humidity at five temperatures: 0℃, 10℃, 22℃, 27℃, and 40℃. One hundred100 random draws from the posterior distribution are shown at each temperature to visualize uncertainty. Line and point colors indicate temperature. (b) Predicted half-life above the ERH as a function of quasi-equilibrium concentration factor. Points and lines as in a, but only solution (above ERH) conditions are shown. (c) Heatmap showing posterior median predicted half-lives as a function of temperature and relative humidity. Posterior median estimated half-lives for human coronaviruses from our study and from the literature plotted on top using the same color map (see also *Appendix 1—table 3* and *Figure 3—figure supplement 1*). Shape indicates virus; measurements from our own group are shown slightly larger with a slightly thicker outline. Points of identical temperature and humidity are nudged slightly to avoid direct overplotting. (d) Relative within-study mechanistic model predictions (x-axis, see Appendix, Relative predictions) compared to simple regression measurements (y-axis) for human coronavirus half-lives. Points show posterior median for measured (horizontal) or predicted (vertical) half-lives and lines show a 68% (thick) and 95% (thin) credible interval. Shape indicates virus; datapoints come from studies in the literature for which there were measurements at at least two temperature and/or humidity conditions for the same virus and experimental material (e.g. plastic, steel, bulk medium).

The online version of this article includes the following source data and figure supplement(s) for figure 3:

**Source data 1.** Predicted and measured half-lives (posterior medians and credible intervals) for within-study relative predictions shown in d.
**Figure supplement 1.** Absolute mechanistic model predictions compared simple regression estimates of half-lives, as in c, but plotted as in a and d.
**Figure supplement 1—source data 1.** Predicted and measured half-lives (posterior medians and credible intervals) for absolute predictions shown in c and in *Figure 3—figure supplement 1*.

important determinant of superspreading risk. Our results provide a mechanistic explanation for the many observed SARS-CoV-2 superspreading events in cool indoor environments such as food processing plants (*Dyal, 2020*; *Günther et al., 2020*; *Pokora et al., 2020*) and hockey rinks (*Atrubin et al., 2020*; *McNabb and Ries, 2020*), where the typical air temperature is around 10℃, or in dry indoor environments such as long-distance flights (*Khanh et al., 2020*; *Jayaweera et al., 2020*). Conversely, our results imply that the relative rarity of outdoor SARS-CoV-2 transmission clusters is not readily explained by temperature and humidity effects, since these conditions outdoors during temperate zone winters should be favorable for the virus. Instead, increased ventilation (*Prather et al., 2020*) and UV light inactivation (*Ratnesar-Shumate et al., 2020*) may be more important than the effects of temperature and humidity outdoors. In contrast, typical climate-controlled conditions indoors (moderate temperature and low humidity) are favorable for virus stability, and specialized conditions such as those found in food processing plants even more so. Our results highlight the importance of proper personal protective equipment and improved ventilation for protecting workers, particularly in cold indoor settings, and the general transmission risks associated with indoor gatherings.

The effects of temperature and humidity we observe in our data and model are relevant both to fomite and to airborne transmission. Prior work has shown that virus decay as a function of RH is similar in droplets on surfaces and suspended aerosols (*Lin and Marr, 2020*; *Kormuth et al., 2018*). Numerous studies of smaller deposited droplets (*Prussin et al., 2018*) or aerosols (*Benbough, 1971*; *Yang et al., 2012*; *Ijaz et al., 1985*) have reported similar qualitative patterns to those we report, with increased decay rates at high temperatures and a U-shaped effect of RH. Furthermore, surface stability can matter for aerosol transmission risk, since small particles containing infectious virions can be re-suspended from surfaces and inhaled (*Asadi et al., 2020*). Re-suspension is further enhanced by procedures such as high-pressure washing, which is common in food processing plants. While the relative contributions of aerosol and fomite transmission to the epidemiology of SARS-CoV-2 continue to be investigated (*Ong et al., 2020*; *Cai et al., 2020*), our results indicate that cold situations present elevated transmission risks for either mode, especially if air is either dry or very humid. It has been speculated, for instance, that chilled or frozen foods might allow for rare but impactful long-range fomite transmission (*Han et al., 2020*). Our results show that this is conceivable, as there is good empirical and mechanistic support for prolonged virus viability at very low temperatures.

Environmental stability is not the only mechanism by which temperature and humidity affect respiratory virus transmission. Very hot or cold conditions outdoors can lead people to spend more time indoors, where transmission risks are heightened due to poor ventilation. Low-humidity environments can dry out human airways and thus impair defenses against respiratory viruses (*Kudo et al., 2019*). Ambient humidity also determines the size distribution of aerosols in the environment, again

by affecting evaporation rates. Smaller aerosols settle to the ground more slowly (*Marr et al., 2019*), which could facilitate transmission.

At low RH, humidity effects on inactivation, immunity, and settling may compound each other: all increase transmission risk. At high RH, reduced inactivation could promote transmission, but improved immune defenses and faster settling could hinder it, so the net effect on transmission is less clear.

Still, temperate zone winters increase transmission of many respiratory viruses (*Lofgren et al., 2007*). Individuals spend increased time indoors in heated buildings. Ventilation is often poor, as windows are kept closed to make heating efficient. Air in heated buildings is typically very dry; this improves virus stability and weakens immune defenses. Policymakers should consider ventilating and humidifying essential indoor spaces to reduce transmission risk. Other mitigation measures such as indoor masking may likewise be even more crucial during winter. Indoor spaces in which individuals cannot be masked, such as bars and restaurants, remain particular cause for concern.

Several analyses have projected that SARS-CoV-2 transmission will likewise be faster in temperate zone winters (*Neher et al., 2020*; *Kissler et al., 2020*; *Baker et al., 2020*). Major seasonal or climate-mediated mitigation of SARS-CoV-2 spread was not evident during the northern hemisphere's spring and summer (*Carlson et al., 2020*; *Poirier et al., 2020*). This was expected, since population susceptibility and epidemic control measures can be more important than seasonality in an early pandemic context (*Baker et al., 2020*). Thus, the fact that temperate zone summers did not eliminate transmission should not have led to false confidence that temperate zone winters would not promote it. Winter surges in cases, hospitalizations, and deaths across the northern hemisphere may have been driven in part by behavioral, immunological, or virological seasonality.

Our work has implications for the study of virus environmental stability and seasonality more broadly. Whether absolute or relative humidity is more important for influenza stability has been a matter of debate (*Shaman et al., 2010*; *Marr et al., 2019*). The answer has proved elusive because it is difficult to disentangle the effects of humidity from those of temperature. Our mechanistic model permits principled dis-aggregation of those effects, and reveals a strong effect of relative humidity even after accounting for the effects of temperature.

There may thus exist general principles that govern virus inactivation across enveloped viruses, and perhaps even more broadly. Similar empirical patterns of temperature and humidity dependence to what we measured, and modeled, for SARS-CoV-2 have been observed for other important viruses. In particular, the U-shaped dependence of inactivation on RH has been reported for animal coronaviruses (*Songer, 1967*; *Casanova et al., 2010*), as well as for influenza viruses, paramyxoviruses, rhabdoviruses, and retroviruses (*Yang et al., 2012*; *Benbough, 1971*; *Prussin et al., 2018*; *Webb et al., 1963*), suggesting the existence of a shared mechanism for the effect of humidity across enveloped RNA viruses. Some enveloped DNA viruses such as herpesviruses and poxviruses (*Songer, 1967*; *Webb et al., 1963*) and some encapsulated viruses such as polioviruses (*de Jong and Winkler, 1968*; *Songer, 1967*) also show similar empirical behavior. Experiments have found that heat treatment of viruses reduces infectivity principally by degrading surface proteins (*Wigginton et al., 2012*), lending further support to a chemical model of environmental virus inactivation.

Individual enveloped viruses may be more or less stable than SARS-CoV-2 while still obeying our model's basic principle: increased heat and concentration lead to faster inactivation. The values of model parameters ($E_a$, $A_{eff}$, $A_{sol}$) may change while the mechanistic model itself remains valid. The data from our own group and from the literature on MERS-CoV is suggestive in this regard: our model predictions using SARS-CoV-2 parameters slightly overestimate the stability of MERS-CoV, but correctly predict the pattern of temperature and humidity effects (*Figure 3—figure supplement 1*).

Similarly, it is striking that our model for Arrhenius-like temperature dependence works well with a single estimated activation energy across the effloresced and solution regimes for our SARS-CoV-2 experiments and for experiments on a range of coronaviruses conducted in different conditions by other investigators. This suggests that the rate-limiting step in coronavirus inactivation may not necessarily depend on the exact inactivating reactant. We propose one simple potential mechanism for how this could be so: if inactivation depends on disruption of the virion once it has formed a complex with some inactivating reactant, the activation energy for that disruption event could depend

mainly on the chemical properties of the virion itself (see Appendix, Interpretation of the single activation energy).

We discuss additional practical implications for the empirical study of virus environmental stability in the Appendix (Methodological implications for experimental studies on virus stability).

Despite years of research on virus stability as a function of temperature and humidity and plausible hypotheses about the underlying chemistry, proposed mechanisms have lacked explicit quantitative support. By encoding the underlying chemistry into a mathematical model and estimating parameters using modern computational techniques, we provide such support, with critical insights for the control of an ongoing pandemic. Our empirical results provide mechanistic insight into transmission risks associated with cold and climate-controlled indoor settings, while our modeling work allows for explicit quantitative comparison of the aerosol and fomite risks in different environments, and suggests that simple, general mechanisms govern the viability of enveloped viruses: hotter, more concentrated solutions are favorable to chemical reactions—and therefore unfavorable to viruses.

## Materials and methods

### Laboratory experiments

#### Viruses and titration

We used SARS-CoV-2 strain HCoV-19 nCoV-WA1-2020 (MN985325.1; *Holshue et al., 2020*) for this study. We quantified viable virus by end-point titration on Vero E6 cells as described previously (*Fischer et al., 2020*; *van Doremalen et al., 2020*), and inferred posterior distributions for titers and exponential decay rates directly from raw titration data using Bayesian statistical models (see Statistical analyses and mathematical modeling, below).

#### Virus stability experiment

We measured virus stability on polypropylene (ePlastics, reference PRONAT.030X24X47S/M) as previously described (*van Doremalen et al., 2020*). We prepared a solution of Dulbecco's Modified Eagle Medium (DMEM, a common cell culture medium) supplemented with 2 mM L-glutamine, 2% fetal bovine serum, and 100 units/mL penicillin/streptomycin, and containing $10^5$ TCID$_{50}$/mL SARS-CoV-2. Polypropylene disks were autoclaved for decontamination prior to the experiment. We then placed 50 µL aliquots of this SARS-CoV-2 suspension onto the polypropylene disks under nine environmental conditions: three RH (40%, 65%, and 85%) at each of three temperatures (10°C, 22°C, and 27°C). These controlled environmental conditions were produced in incubators (MMM Group CLIMA-CELL and Caron model 6040) with protection from UV-B or UV-C exposure. We prepared 216 disks corresponding to three replicates per eight post-deposition time-points (0, 1, 4, 8, and 24 hr, then daily for 4 days) for the nine conditions. At each time-point, samples were collected by rinsing the disks with 1 mL of DMEM and stored at −80°C until titration.

#### Evaporation experiment

We measured the evaporation kinetics of suspension medium under the same temperature and humidity conditions as the virus stability experiments. We placed 50 µL aliquots of supplemented DMEM onto polypropylene disks in a Electro-Tech Systems 5518 environmental chamber. The polypropylene disks were rinsed three times 1M sulfuric acid, ethanol and DI H$_2$O respectively before use. We measured medium mass $m(t)$ every 5 min for up to 20 hr or until a quasi-equilibrium was reached using a micro-balance (Sartorius MSE3.6P-000-DM, readability 0.0010 mg). The chamber of the micro-balance was half-opened to keep air circulating with the environmental chamber. The flow entering the balance chamber decreased the balance accuracy to around 0.01 mg. We measured initial droplet mass ($m(0)$) and final droplet mass ($m(\infty)$) under closed-chamber conditions to increase accuracy.

### Statistical analyses and mathematical modeling

We quantified the stability of SARS-CoV-2 under different conditions by estimating the decay rates of viable virus titers. We inferred individual titers using a Bayesian model we have previously described (*Gamble et al., 2021*). Briefly, the model treats titration well infection as a Poisson single-

hit process. We inferred raw exponential decay rates by modifying a previously-described simple regression model (*Gamble et al., 2021*) to account for the evaporation phase. See the Appendix (Empirical virus decay estimation) for model description.

We estimated parameters of our mechanistic models by predicting titers based on those models and then applying the same Poisson single-hit observation process to estimate parameters from the data. See Appendix (Mechanistic model estimation) for a complete description, including model priors.

We estimated evaporation rates and corresponding drying times by modeling mass loss for each environmental condition $i$ as linear in time at a rate $\beta_i$ until the final mass $m(\infty)$ was reached. See Appendix (Modeling of medium evaporation and Evaporation model fitting) for a full description, including model priors.

We drew posterior samples using Stan (*Stan Development Team, 2018*), which implements a No-U-Turn Sampler (a form of Markov Chain Monte Carlo), via its R interface RStan (*Stan Development Team, 2016*). We inferred all parameters jointly (e.g. evaporation parameters and mechanistic model parameters were inferred in light of each other).

## Meta-analysis

To test the validity of our model beyond the measured environmental conditions (i.e. beyond 10–27° C and 40–85% RH), we compiled data from 11 published studies on human coronaviruses, including SARS-CoV-2, SARS-CoV-1, MERS-CoV, HCoV-OC43, and HCoV-299E, under 17 temperature-RH conditions. We generated estimates of half-lives and uncertainties (*Appendix 1—table 3*) and compared those estimates to the half-lives predicted by the mechanistic model parametrized from our SARS-CoV-2 data. As data on evaporation kinetics were not available, we estimated a unique half-life for each experimental condition, covering both the evaporation and quasi-equilibrium phases. As virus decay during the evaporation phase is expected to be minimal, and the evaporation phase to be short, the estimated half-life can be used as a proxy for the quasi-equilibrium half-life. The complete data selection, extraction and analysis process is detailed in the Appendix (Meta-analysis of human coronavirus half-lives).

We also included data from SARS-CoV-1 and MERS-CoV collected by our group during previous studies (*van Doremalen et al., 2020*). Those data were collected at 22°C and 40% RH on polypropylene using the protocol described previously (*van Doremalen et al., 2020*) and similar to the one used to collect the SARS-CoV-2 data. SARS-CoV-1 strain Tor2 (AY274119.3) (*Marra et al., 2003*) and MERS-CoV strain HCoV-EMC/2012 (*Zaki et al., 2012*) were used for these experiments. We calculated half-lives for evaporation and quasi-equilibrium phases using the same analysis pipeline used for SARS-CoV-2 (Appendix, Empirical virus decay estimation). These data were used only for out-of-sample prediction testing. We used the obtained evaporation phase half-lives as proxies for the half-life at 100% RH, as with SARS-CoV-2. See Appendix for a figure showing model fits (*Appendix 1—figure 23*) and a table of estimated half-lives (*Appendix 1—table 4*).

## Visualization

We created plots in R using ggplot2 (*Wickham, 2016*), ggdist (*Kay, 2020a*), and tidybayes (*Kay, 2020b*), and created original schematics using BioRender.com.

## Acknowledgements

We thank *eLife* editors W Garrett and CB Ogbunu and two anonymous reviewers for valuable comments on the manuscript. This research was supported by the Intramural Research Program of the National Institute of Allergy and Infectious Diseases (NIAID), National Institutes of Health (NIH). DHM was supported by the US National Science Foundation (CCF 1917819). JOL-S and AG were supported by the Defense Advanced Research Projects Agency DARPA PREEMPT #D18AC00031 and the UCLA AIDS Institute and Charity Treks, and JOL-S was supported by the US National Science Foundation (DEB-1557022), the Strategic Environmental Research and Development Program (SERDP, RC-2635) of the US QH, LCM, and PJV were supported by the US National Science Foundation (CBET-1705653, CBET-2029911). The content of the article does not necessarily reflect the position or the policy of the US government, and no official endorsement should be inferred.

## Additional information

### Funding

| Funder | Grant reference number | Author |
|---|---|---|
| National Science Foundation | CCF 1917819 | Dylan H Morris |
| Defense Advanced Research Projects Agency | D18AC00031 | Amandine Gamble<br>James O Lloyd-Smith |
| UCLA AIDS Institute and Charity Treks | | Amandine Gamble<br>James O Lloyd-Smith |
| National Institute of Allergy and Infectious Diseases | Intramural Research Program | Kwe Claude Yinda<br>Trenton Bushmaker<br>Robert J Fischer<br>M Jeremiah Matson<br>Neeltje Van Doremalen<br>Vincent J Munster |
| National Science Foundation | DEB-1557022 | James O Lloyd-Smith |
| Strategic Environmental Research and Development Program | RC-2635 | James O Lloyd-Smith |
| National Science Foundation | CBET-1705653 | Qishen Huang<br>Peter J Vikesland<br>Linsey C Marr |
| National Science Foundation | CBET-2029911 | Qishen Huang<br>Peter J Vikesland<br>Linsey C Marr |

The funders had no role in study design, data collection and interpretation, or the decision to submit the work for publication.

### Author contributions

Dylan H Morris, Conceptualization, Software, Formal analysis, Validation, Investigation, Visualization, Methodology, Writing - original draft, Writing - review and editing, Led development of mechanistic model; Kwe Claude Yinda, Conceptualization, Investigation, Methodology, Writing - original draft, Writing - review and editing, Led empirical measurement work; Amandine Gamble, Conceptualization, Data curation, Formal analysis, Validation, Investigation, Methodology, Writing - original draft, Writing - review and editing, Led meta-analysis; Fernando W Rossine, Formal analysis, Investigation, Methodology, Writing - review and editing; Qishen Huang, Trenton Bushmaker, Robert J Fischer, M Jeremiah Matson, Investigation, Writing - review and editing; Neeltje Van Doremalen, Investigation, Methodology, Writing - review and editing; Peter J Vikesland, Linsey C Marr, Formal analysis, Supervision, Funding acquisition, Validation, Methodology, Writing - review and editing; Vincent J Munster, Conceptualization, Resources, Methodology, Project administration, Writing - review and editing; James O Lloyd-Smith, Conceptualization, Formal analysis, Supervision, Funding acquisition, Validation, Investigation, Writing - original draft, Writing - review and editing

### Author ORCIDs

Dylan H Morris https://orcid.org/0000-0002-3655-406X
Kwe Claude Yinda https://orcid.org/0000-0002-5195-5478
Amandine Gamble https://orcid.org/0000-0001-5430-9124
Qishen Huang https://orcid.org/0000-0002-8493-5799
Trenton Bushmaker https://orcid.org/0000-0002-2161-4808
Robert J Fischer https://orcid.org/0000-0002-1816-472X
M Jeremiah Matson https://orcid.org/0000-0003-3404-1380
Neeltje Van Doremalen http://orcid.org/0000-0003-4368-6359
Peter J Vikesland https://orcid.org/0000-0003-2654-5132

Linsey C Marr (iD) https://orcid.org/0000-0003-3628-6891
James O Lloyd-Smith (iD) https://orcid.org/0000-0001-7941-502X

**Decision letter and Author response**
Decision letter https://doi.org/10.7554/eLife.65902.sa1
Author response https://doi.org/10.7554/eLife.65902.sa2

## Additional files

**Supplementary files**
• Transparent reporting form

**Data availability**

All code and data needed to reproduce results and figures is archived on Github (https://github.com/dylanhmorris/sars-cov-2-temp-humidity/; copy archived at https://archive.softwareheritage.org/swh:1:rev:d80dd0132be738753f1a77c99ce280219dc5afba) and on Zenodo (https://doi.org/10.5281/zenodo.4093264), and licensed for reuse, with appropriate attribution/citation, under a BSD 3-Clause Revised License. This includes all original data generated in the experiments and all data collected and used for meta-analysis.

The following dataset was generated:

| Author(s) | Year | Dataset title | Dataset URL | Database and Identifier |
|---|---|---|---|---|
| Morris DH | 2021 | Code and data for Mechanistic theory predicts the effects of temperature and humidity on inactivation of SARS-CoV-2 and other enveloped viruses | https://doi.org/10.5281/zenodo.4093264 | Zenodo, 10.5281/zenodo.4093264 |

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

## Appendix 1

### Mechanistic inactivation model interpretation
#### Interpretation of the single activation energy

We observe in the main text that a single activation energy explains the data well across the efflo-resced and solution regimes (*Appendix 1—figure 1*).

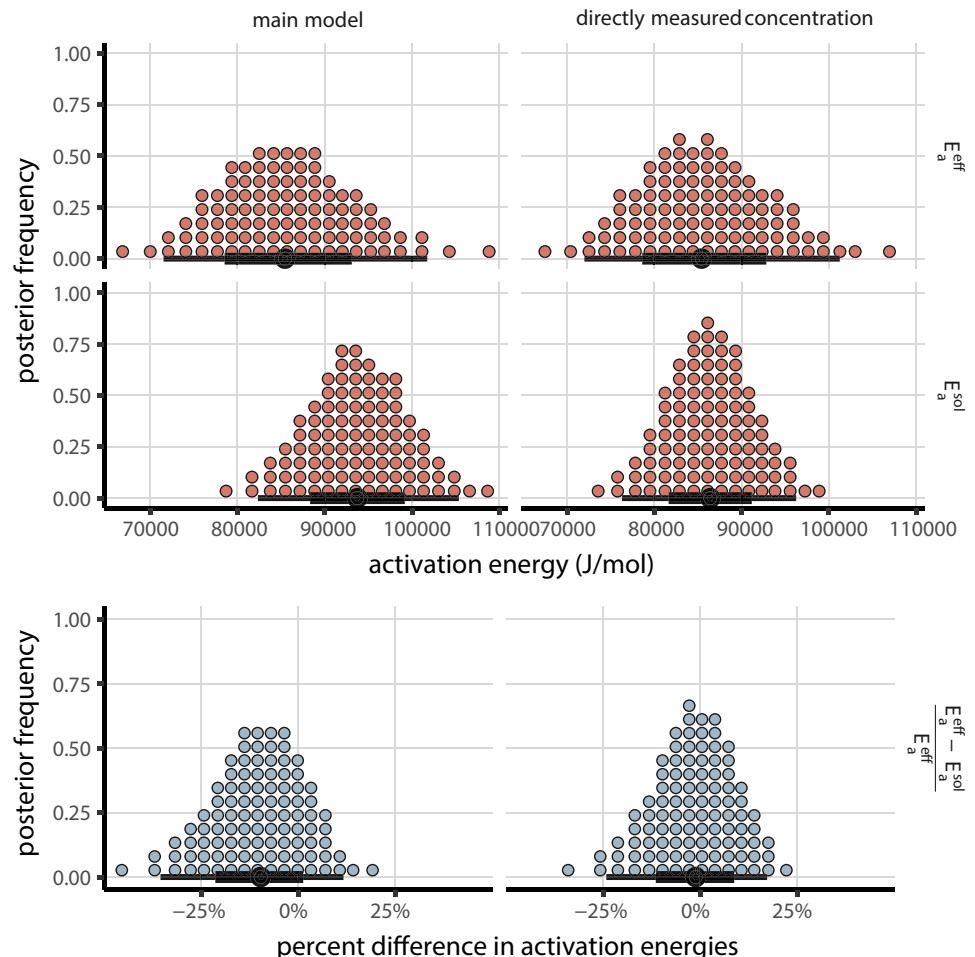

**Appendix 1—figure 1.** Posterior distributions for activation energies below ($E_a^{\text{eff}}$) and above ($E_a^{\text{sol}}$) the ERH, and the percentage difference between them $\left(\frac{E_a^{\text{eff}} - E_a^{\text{sol}}}{E_a^{\text{eff}}}\right)$, 4-parameter model version. Main model fit shown at left, model fit with directly-measured concentration shown at right. Distributions are visualized as quantile dotplots (*Kale et al., 2020*); 100 representative dots are shown for each parameter. Black circle below shows posterior median, bars show 68% (thick) and 95% (thin) credible intervals.

Moreover, our estimate is consistent with activation energies observed for other RNA viruses (*Rowell and Dobrovolny, 2020*). Our median [95% credible interval] $E_a$ estimate from the main model, $9.10 \times 10^4 \text{ J mol}^{-1}$ [$8.21 \times 10^4$, $1.01 \times 10^5$], falls squarely within the range of literature esti-mates (approximately $6.00 \times 10^4$ to $2.40 \times 10^5$) (*Rowell and Dobrovolny, 2020*).

These observations raise the question of whether the actual inactivating reaction is identical in the effloresced and solution regimes, in different media, and for different viruses. But at least for a given virus or family of viruses, it is possible for virus inactivation reactions to have the same activa-tion energy even if different media or different environments imply a different inactivating reactant. Plausible routes of chemical virus inactivation include conformational changes in virion proteins,

disruption of the virus capsid (*Wigginton et al., 2012*), and disruption of the virus envelope (*Yang and Marr, 2012*). These may occur via a two-step reaction:

$$\text{viable virion} + \text{external reactant} \leftrightarrow \text{intermediate product} \rightarrow \text{inactivated virion} \tag{4}$$

If the second step is rate-limiting, then the overall reaction kinetics are first order and the measured activation energy will reflect the $E_a$ for that step. This energy could easily depend only on the virus proteins or envelope and not on the external reactant.

## Two-step reactions can produce first-order kinetics proportional to concentration

Provided the external reactant concentration $[r]$ is not meaningfully depleted, a two-step inactivation reaction of this form would still imply a linear dependence of inactivation rate on concentration of external reactant, and thus a linear dependence on solution concentration as postulated in our model (*Equation 3*). Below we describe a minimal two-step reaction mechanism that is consistent with these observations.

We denote the concentration of viable virus by $[v_v]$, the concentration of inactivated virus by $[v_i]$, and the concentration of intermediate product by $[x]$. We denote the rate constants for the forward and backward first-step reactions by $k_1^+$ and $k_1^-$ and the rate constant for the second-step reaction by $k_2^+$. We have:

$$\begin{aligned}
\frac{\mathrm{d}[v_v]}{\mathrm{d}t} &= -k_1^+[r][v_v] + k_1^-[x] \\
\frac{\mathrm{d}[x]}{\mathrm{d}t} &= k_1^+[r][v_v] - k_1^-[x] - k_2^+[x] \\
\frac{\mathrm{d}[v_i]}{\mathrm{d}t} &= k_2^+[x]
\end{aligned} \tag{5}$$

By assumption, the first step in the reaction is fast relative to the second. The intermediate product $[x]$ should therefore reach a quasi-equilibrium value $[\bar{x}]$. We solve for it by setting $\frac{\mathrm{d}[x]}{\mathrm{d}t}$ and neglecting the smaller $-k_2^+$ term:

$$[\bar{x}] = [r]\frac{k_1^+}{k_1^-}[v_v] \tag{6}$$

Substituting $[\bar{x}]$ for $[x]$ into $\frac{\mathrm{d}[v_i]}{\mathrm{d}t}$, it follows that virus inactivation obeys first-order kinetics proportional to the external reactant concentration $[r]$:

$$\frac{\mathrm{d}[v_i]}{\mathrm{d}t} = [r]k_2^+\frac{k_1^+}{k_1^-}[v_v] \tag{7}$$

## Interpretation of the asymptotic reaction rates

We also observe that the pre-exponential factor (asymptotic high temperature reaction rate) is somewhat but not substantially greater in the effloresced regime than in the solution regime ($A_{\mathrm{eff}} > A_{\mathrm{sol}}$). Since $A_{\mathrm{sol}}$ is modulated by $\frac{[S_{\mathrm{eq}}]}{[S_0]}$, this implies that reaction rates in the effloresced crystals (which we assume occur at the same rate for all sub-ERH ambient humidities) are faster than reactions at 100% RH, but not as fast as at humidities slightly above the ERH, such as 65% (*Figure 2—figure supplement 2*, *Appendix 1—table 1*).

This empirical result is plausible. Below the ERH, reactants are in closer proximity, but also less mobile: modeled as a quasi-solution, there is a higher reactant concentration but also a lower diffusion coefficient. It is thus plausible that the effective rate of potentially reactive collisions for a given temperature could be greater than the rate in dilute solution at 100% RH, but substantially lower than the rate in more concentrated solution at 65% RH.

## Interpretation of the transition in inactivation rate at the ERH

Since $A_{\text{eff}}$ and $A_{\text{sol}}$ are estimated separately in our model, there is a discontinuity in the inactivation rate at the ERH (*Figure 3a*). In reality, there may be a more continuous transition. Molecular interactions may interpolate between the fully-effloresced and fully-solution states, resulting in a continuous phase transition-like behavior. But as multiple measurements close to the ERH on both sides of it would be required to characterize this behavior conclusively, it is beyond the scope of our study. We therefore allow a discontinuity at the ERH.

## Model parameter estimate tables

Here we provide tables of key parameter estimates for the mechanistic model of SARS-CoV-2 inactivation as a function of temperature and humidity. *Appendix 1—table 1* shows estimates obtained using a fitted curve relating RH to concentration factor, as in the main text. *Appendix 1—table 2* shows estimates obtained using concentration factors directly measured in evaporation experiments. See *Figure 2—figure supplement 2* for visualizations of these parameter estimates.

**Appendix 1—table 1.** Parameter estimates for the mechanistic model of SARS-CoV-2 inactivation as a function of temperature and humidity, using a fitted curve relating RH to concentration factor, as in the main text.
Estimates are reported as posterior median and the middle 95% credible interval.

| Parameter | Median | 2.5 % | 97.5 % | Unit |
|---|---|---|---|---|
| $A_{\text{eff}}$ | $6.15 \times 10^{14}$ | $1.64 \times 10^{13}$ | $3.02 \times 10^{16}$ | $h^{-1}$ |
| $A_{\text{sol}}$ | $2.51 \times 10^{13}$ | $6.31 \times 10^{11}$ | $1.34 \times 10^{15}$ | $h^{-1}$ |
| $E_a$ | $9.10 \times 10^{4}$ | $8.21 \times 10^{4}$ | $1.01 \times 10^{5}$ | $J\,mol^{-1}$ |

**Appendix 1—table 2.** Parameter estimates for the mechanistic model of SARS-CoV-2 inactivation as a function of temperature and humidity, using concentration factors directly-measured in evaporation experiments.
Estimates are reported as posterior median and the middle 95% credible interval.

| Parameter | Median | 2.5 % | 97.5 % | Unit |
|---|---|---|---|---|
| $A_{\text{eff}}$ | $8.70 \times 10^{13}$ | $3.48 \times 10^{12}$ | $2.31 \times 10^{15}$ | $h^{-1}$ |
| $A_{\text{sol}}$ | $3.43 \times 10^{12}$ | $1.27 \times 10^{11}$ | $9.64 \times 10^{13}$ | $h^{-1}$ |
| $E_a$ | $8.62 \times 10^{4}$ | $7.82 \times 10^{4}$ | $9.42 \times 10^{4}$ | $J\,mol^{-1}$ |

## Mechanistic modeling of evaporation and concentration

To measure the solute concentration factor over time and to determine when droplets reached evaporative quasi-equilibrium (i.e. evaporation became slow enough that concentration factor could be treated as a constant), we quantified evaporation of the suspension medium on polypropylene plastic (without virus) at the tested temperature and humidity combinations (Materials and methods; *Figure 1—figure supplement 3*).

To extrapolate to unobserved relative humidities, we estimated the quasi-equilibrium solute concentration factor $\frac{[S_{eq}]}{[S_0]}$ as a function of relative humidity $h$.

The mathematical modeling we describe in this section is not central to our mechanistic model of how temperature and humidity affect virus inactivation. Rather, it is an attempt to conduct principled extrapolation to unobserved conditions. We do not attempt a general or fully mechanistic model of the relationship between relative humidity and quasi-equilibrium concentration factor, as in real-world conditions this will depend on the chemistry of the human fluids in which virions are found and its interactions with a non-ideal environment; this is an important avenue for future research. For our

purposes here, a semi-mechanistic characterization of the increase in concentration factor with decreased relative humidity (up to the ERH) suffices.

Similarly, our analysis here allows us to distinguish the evaporation and quasi-equilibrium phases when doing inference. This matters because the time to reach quasi-equilibrium will vary in real conditions. It was prolonged in some of our experiments because we used large droplets; in everyday scenarios, it may vary from near-instant for small respiratory droplets and aerosols produced in speech to somewhat longer for large droplets produced by a sneeze.

## Solute concentration factor

The concentration factor as a function of time $\frac{[S(t)]}{[S_0]}$ is equal to the ratio of the initial mass of water $w(0)$ (before evaporation begins) to the current mass of water $w(t)$. We measured total masses $m(t)$, not masses of water, but assuming that the mass of solutes, $s$, is conserved:

$$\frac{[S(t)]}{[S_0]} = \frac{w(0)}{w(t)} = \frac{m(0) - s}{m(t) - s} \tag{8}$$

and so:

$$\frac{[S_{eq}]}{[S_0]} = \frac{w(0)}{w(\infty)} = \frac{m(0) - s}{m(\infty) - s} \tag{9}$$

In order to predict decay rates at unobserved relative humidities, we fit a semi-mechanistic function to the measured concentration factors to predict $\frac{[S_{eq}]}{[S_0]}$ as a function of fractional relative humidity $h$. We begin with the observation (see Relationship between concentration factor and solute molar fraction for a derivation) that if $X(\infty)$ is the molar fraction of solutes in the solution at quasi-equilibrium and $X(0)$ is the initial molar fraction of solutes, then:

$$\frac{[S_{eq}]}{[S_0]} = \frac{1 - X(0)}{X(0)} \frac{X(\infty)}{1 - X(\infty)} \tag{10}$$

We denote the initial ratio of the molar fractions by $r(0) = \frac{X(0)}{1 - X(0)}$.

The final molar ratio $r(\infty) = \frac{X(\infty)}{1 - X(\infty)}$ depends on the fractional relative humidity $h$. We approximate this relationship by a flexible two-parameter function:

$$r(\infty) = \frac{X(\infty)}{1 - X(\infty)} = \left( \frac{-\ln(h)}{\alpha_s} \right)^{\frac{1}{\alpha_c}} \tag{11}$$

Combining yields:

$$\frac{[S_{eq}]}{[S_0]} = \frac{1}{r(0)} \left( \frac{-\ln(h)}{\alpha_s} \right)^{\frac{1}{\alpha_c}} \tag{12}$$

The estimated parameters $\alpha_c, \alpha_s > 0$ reflect deviations of our solute mixture from ideal behavior ($\alpha_c = \alpha_s = 1$). We derived this approximate expression from chemical theory; see Derivation of approximate functional form for the quasi-equilibrium solute concentration (*Equation 11*) for the derivation. Ideal chemical behavior would imply a linear relationship between $X(\infty)$ and $h$ near $h = 1 : X(\infty) = 1 - h$. This works well for dilute solutions. But it predicts too high a concentration factor at low relative humidities, since it neglects the increasingly strong effects of solutes in preventing evaporation as those solutes become more concentrated. To extrapolate in a worthwhile way, then, we need at least a minimal model of non-ideal evaporative behavior in a concentrated solution. Our simple function fits our data well (*Figure 2—figure supplement 1*).

## Modeling of medium evaporation

In our evaporation experiments, we observed an approximately linear decrease in water mass $w(t)$ over time (*Figure 1—figure supplement 3*), followed by a leveling off at an approximately constant

value (quasi-equilibrium). We therefore approximated the evaporation process with a piecewise linear function:

$$m(t) = \begin{cases} m(0) - \beta t & \beta t < m(0) - m(\infty) \\ m(\infty) & \text{otherwise} \end{cases} \tag{13}$$

As noted above (see Solute concentration factor), we assumed that solute mass $s$ was conserved, so $w(t) = m(t) - s$. It follows that:

$$w(t) = \begin{cases} m(0) - s - \beta t & \beta t < m(0) - m(\infty) \\ m(\infty) - s & \text{otherwise} \end{cases} \tag{14}$$

This implies that the concentration factor as a function of time is given by:

$$\frac{[S(t)]}{[S_0]} = \frac{w(0)}{w(t)} = \frac{m(0) - s}{m(0) - s - \beta t} \tag{15}$$

Defining $B = \frac{\beta}{w(0)} = \frac{\beta}{m(0) - s}$ yields a normalized form:

$$\frac{[S(t)]}{[S_0]} = \frac{1}{1 - Bt} \tag{16}$$

## Evaporation and quasi-equilibrium phases

In our estimation models, we partitioned virus inactivation into two phases: evaporation and quasi-equilibrium (see Materials and methods). We denote the time to quasi-equilibrium for experiment $i$ by $\tau_i$.

We determined $k_{ev}$ for each inactivation experimental condition based on on the evaporative mass loss rate $\beta_i$ in the corresponding evaporation experiment.

For the simple regression model and the fit of the mechanistic model using only directly-measured concentration, we define $\tau_i$ as the time to reach the measured final total mass $m_i(\infty)$ from the measured initial total mass $m_i(0)$, given the inferred evaporative mass loss rate $\beta_i$:

$$\tau_i = \frac{m_i(0) - m_i(\infty)}{\beta_i} \tag{17}$$

For the main fit of the mechanistic model, in which we use a fitted curve relating RH to $\frac{[S_{eq}]}{[S_0]}$, we partition the phases not based on $\tau_i$ but rather based on the time $\bar{\tau}_i$ to reach the predicted quasi-equilibrium concentration factor $\frac{[S_{eq}]}{[S_0]}_i$ given the inferred $B_i = \frac{\beta_i}{w_i(0)}$:

$$\bar{\tau}_i = \frac{1 - \frac{1}{\frac{[S_{eq}]}{[S_0]}_i}}{B_i} \tag{18}$$

Note that this relation also holds for directly-measured concentration. Letting $\frac{[S_{eq}]}{[S_0]} = \frac{m(0) - s}{m(\infty) - s}$, *Equation 18* simplifies to *Equation 17*.

## Modeling of virus decay dynamics during the evaporation phase

Prior to evaporative quasi-equilibrium or complete efflorescence, virions are in wet conditions, with non-negligible evaporation ongoing. The degree of concentration of that solution $\frac{[S(t)]}{[S_0]}$ changes as a function of time as the solvent (here, suspension medium) evaporates, until a quasi-equilibrium state is reached at $\frac{[S(t)]}{[S_0]} = \frac{[S_{eq}]}{[S_0]}$.

Per *Equation 8*, the concentration factor as a function of time is equal to $\frac{w(0)}{w(t)}$.

The inactivation rate during that evaporation phase, which we denote by $k_{ev}$, is then a function of time $k_{ev}(t)$:

$$k_{\text{ev}}(t) = \frac{w(0)}{w(t)} A_{\text{sol}} \exp\left(-\frac{E_a}{RT}\right) \tag{19}$$

Letting $v(t)$ denote the quantity of viable virus, inactivation kinetics will then proceed according to the differential equation:

$$\frac{\mathrm{d}v}{\mathrm{d}t} = -k_{\text{ev}}(t)v \tag{20}$$

We define $k_0 = k_{\text{sol}}(0) = A_{\text{sol}} \exp\left(-\frac{E_a}{RT}\right)$ and apply our linear evaporation model from **Equation 15**:

$$\frac{\mathrm{d}v}{\mathrm{d}t} = -k_{\text{ev}}(t)v = -\frac{k_0}{1 - \left(\frac{\beta}{w(0)}\right)t}v = -\frac{k_0}{1 - Bt}v \tag{21}$$

Solving yields:

$$v(t) = v(0)\exp\left(\frac{k_0}{B}\ln(1 - Bt)\right) = v(0)(1 - Bt)^{(k_0/B)} \tag{22}$$

subject to the constraint that $Bt < 1$, which is always satisfied for $t \leq \tau$, under the assumption that some non-zero amount of water remains at quasi-equilibrium.

Since virus titers are typically measured in $\log_{10}$ units, it is useful to have this expression in those terms:

$$\log_{10}(v(t)) = \log_{10}(v_0) + \frac{k_0}{B}\log_{10}(1 - Bt) \tag{23}$$

## Relationship between concentration factor and solute molar fraction (*Equation 10*)

Under the assumption that mass of solute does not change, all mass change reflects loss or gain of solvent. This mass change translates directly into increased or decreased concentration, and allows us to compute the estimated concentration factor as a function of time, $\frac{[S(t)]}{[S_0]}$, based on our evaporation experiments.

If we have $N_w(t)$ moles of solvent versus an initial value of $N_w(0)$ and a constant number $N_s$ of solute, then following a similar reasoning as in **Equation 8**:

$$\frac{[S(t)]}{[S_0]} = \frac{N_w(0)}{N_w(t)} \tag{24}$$

If $X(t)$ is the mole fraction of solutes in the solution, $N_w(t) = (1 - X(t))N(t)$ and $N_s = X(t)N(t)$ where $N(t) = N_w(t) + N_s$. It follows that the ratio of moles is the ratio of the mole fractions:

$$\frac{N_w(t)}{N_s} = \frac{1 - X(t)}{X(t)} \tag{25}$$

Since $N_s$ does not change:

$$\frac{N_w(0)}{N_s} = \frac{(1 - X(0))}{X(0)} \tag{26}$$

Hence:

$$\frac{[S(t)]}{[S_0]} = \frac{N_w(0)}{N_w(t)} = \frac{N_w(0)/N_s}{N_w(t)/N_s} = \frac{\frac{1 - X(0)}{X(0)}}{\frac{1 - X(t)}{X(t)}} = \left(\frac{1 - X(0)}{X(0)}\right)\frac{X(t)}{1 - X(t)} \tag{27}$$

and therefore:

$$\frac{[S_{\text{eq}}]}{[S_0]} = \left(\frac{1-X(0)}{X(0)}\right)\frac{X(\infty)}{1-X(\infty)} \tag{28}$$

## Derivation of approximate functional form for the quasi-equilibrium solute concentration (*Equation 11*)

To compute $\frac{[S_{\text{eq}}]}{[S_0]}$ as a function of fractional relative humidity $h$, we need an expression for the ratio of the quasi-equilibrium solute mole fraction $X(\infty)$ to the quasi-equilibrium solvent mole fraction $1 - X(\infty)$ as a function of $h$.

An evaporating aqueous solution reaches equilibrium with the ambient air when the ambient relative humidity is equal to the water activity $a_w$ in the solution:

$$h = a_w \tag{29}$$

For an ideal solution, the water activity would be given by:

$$a_w = 1 - X(\infty) \tag{30}$$

where $X(\infty)$ is the mole fraction of solutes (Raoult's law). In a real solution, this expression must be modified to account for non-ideal behavior.

If there are $n$ species of solute ions and/or molecules present with molar fractions $X_j$, we express this non-ideality in terms of the practical osmotic coefficient $\phi(X_1, ...X_n)$ (*Blandamer et al., 2005*), which is in general a function of the $X_j$:

$$a_w = \exp\left(-\phi\frac{\sum_{j=1}^n X_j}{1-\sum_{j=1}^n X_j}\right) = \exp\left(-\phi\frac{X(\infty)}{1-X(\infty)}\right) \tag{31}$$

Since our medium has a consistent solute formulation and we assume that solutes are conserved, we can treat $\phi$ as a function of the total solute molar fraction $X(t)$. We use the following flexible functional form for $\phi$:

$$\phi = \alpha_s\left(\frac{X}{1-X}\right)^{\alpha_c - 1} \tag{32}$$

With $\alpha_c, \alpha_s > 0$. We define these constrained parameters in terms of unconstrained parameters $c_c$ and $c_s$:

$$\begin{aligned}\alpha_c &= \exp(-c_c)\\ \alpha_s &= \exp(-c_s)\end{aligned} \tag{33}$$

It follows that:

$$a_w = \exp\left[-\alpha_s\left(\frac{X}{1-X}\right)^{\alpha_c}\right] \tag{34}$$

This is a flexible two-parameter function with a number of desirable properties.

- $X = 0$ implies $a_w = 1$ and $X = 1$ implies $a_w = 0$, as should be the case.
- When $c_c, c_s = 0$, the relationship approximates the linear behavior observed in the ideal case, and we have $\phi = 1$ regardless of $X$, reflecting this ideality.
- $c_c < 0$ implies a concave relationship between mole fraction and activity near $a_w = 1$, $c_c > 0$ implies a convex relationship there, and $c_c = 0$ implies a linear relationship.
- Varying $c_s$ controls the steepness of the relationship near $a_w = 1$ while preserving concavity in that region; larger values imply a steeper relationship.
- Empirical $\phi(X)$ functions for important solute components of DMEM, such as NaCl, are monotonically increasing in $X$ over the range of expected equilibrium mole fractions (*Mikhailov et al., 2004*), and thus should be readily approximated by our function.

Using the property that evaporative equilibrium occurs when $a_w = h$, we approximate the ratio $r(\infty)$ of solute to solvent mole fractions at quasi-equilibrium (*Equation 11*) by:

$$r(\infty) = \frac{X(\infty)}{1 - X(\infty)} = \left(\frac{-\ln(h)}{\alpha_s}\right)^{\frac{1}{\alpha_c}}$$

This function readily approximates a number of realistic shapes (*Mikhailov et al., 2004*; *Redrow et al., 2011*) for the relationship between $X$ and $h$, particularly on the interval of interest, between 100% relative humidity and the efflorescence relative humidity (ERH) ($1 \geq h \geq \mathrm{ERH} \approx 0.45$).

This function has simpler approximations to the humidity-molar-ratio relationship as special cases. For instance, $\alpha_c = 1$ implies that $\phi$ does not vary with solute mole fraction $X$ (as happens in ideal solutions).

The main downside of this function is that our $\phi(X)$ is constrained to be be monotonic. It is thus impossible for the relationship between $h$ and $\frac{X}{1-X}$ to have more than one concavity change the range $[0, 1]$. But this is unlikely to be important given that we are mainly interested (and fitting to) the range from the ERH to 100% relative humidity. In fact, an always-concave function readily explains our evaporation data in that range (*Figure 2—figure supplement 1*).

Plugging *Equation 11* into *Equation 10* yields the expression for $\frac{[S_{\mathrm{eq}}]}{[S_0]}$ in terms of the initial solute mole fraction ratio $r(0) = \frac{X(0)}{1-X(0)}$ and the ambient relative humidity $h$ given in *Equation 12*:

$$\frac{[S_{\mathrm{eq}}]}{[S_0]} = \frac{1}{r(0)}\left(\frac{-\ln(h)}{\alpha_s}\right)^{\frac{1}{\alpha_c}}$$

Notice that while quasi-equilibrium concentration factors will depend on both $\alpha_c$ and $\alpha_s$, the ratio of two quasi-equilibrium concentration factors from the same baseline (i.e. $\frac{[S_{\mathrm{eq}}^a]/[S_0]}{[S_{\mathrm{eq}}^b]/[S_0]}$ for two different ambient humiditites $h_a$ and $h_b$) will depend only on $\alpha_c$:

$$\frac{[S_{\mathrm{eq}}^a]/[S_0]}{[S_{\mathrm{eq}}^b]/[S_0]} = \left(\frac{\ln(h_a)}{\ln(h_b)}\right)^{\frac{1}{\alpha_c}} \tag{35}$$

Using *Equation 35* in conjunction with *Equation 3*, one can predict a half-life at one temperature-relative humidity pair from a half-life measured at another, provided all else is equal. We use such an approach to make relative predictions in our meta-analysis (*Figure 3d*). See Relative predictions for details.

## Bayesian estimation models

### Model notation

In the model notation that follows, the symbol ~ denotes that a random variable is distributed according to the given distribution. Normal distributions are parametrized as:

$$\mathrm{Normal(mean, standard\ deviation)}$$

Positive-constrained normal distributions ('Half-Normal') are parametrized as:

$$\mathrm{HalfNormal(mode, standard\ deviation)}$$

For each inactivation experiment (set of temperature humidity conditions for a given virus), there is a corresponding medium evaporation experiment, which in which we measured the evaporation of suspension medium at that same temperature and humidity.

### Titer inference

#### Titer inference model

We inferred individual titers directly from titration well data. We modeled individual positive and negative wells for sample $i$ according to a Poisson single-hit model (*Brownie et al., 2011*). That is,

the number of virions that successfully infect cells within a given well is Poisson distributed with mean:

$$\ln(2)10^{v_i} \tag{36}$$

This expression for the mean derives from the fact that our units are TCID$_{50}$; the probability of a positive well at $v_i = 0$, that is, 1 TCID$_{50}$, is equal to $1 - \exp(-\ln(2) \times 1) = 0.5$.

Let $y_{idk}$ be a binary variable indicating whether the $k^{\text{th}}$ well at dilution factor $d$ (where $d$ is expressed as log$_{10}$ dilution factor) for sample $i$ was positive (so $y_{idk} = 1$ if that well was positive and 0 if it was negative). Under a single-hit process, a well will be positive as long as at least one virion successfully infects a cell.

It follows from **Equation 36** that the conditional probability of observing $y_{idk} = 1$ given a true underlying log$_{10}$ titer $v_i$ is given by:

$$\mathcal{L}(y_{idk} = 1 \mid v_i) = 1 - \exp(-\ln(2) \times 10^{(v_i - d)}) \tag{37}$$

This is simply the probability that a Poisson random variable with mean $\ln(2)10^{(v_i-d)}$ is greater than 0, and $v_i - d$ is the expected concentration of virions, measured in log$_{10}$ TCID$_{50}$, in the dilute sample. Similarly, the conditional probability of observing $y_{idk} = 0$ given a true underlying log$_{10}$ titer $v_i$ is:

$$\mathcal{L}(y_{idk} = 0 \mid v_i) = \exp(-\ln(2) \times 10^{(v_i - d)}) \tag{38}$$

which is the probability that the Poisson random variable is equal to 0.

This gives us our likelihood function, assuming independence of outcomes across wells. Titrated doses introduced to each cell-culture well were of volume 0.1 mL, so we incremented inferred titers by 1 to convert to units of log$_{10}$ TCID$_{50}$/mL.

## Titer inference model prior distributions

We assigned a weakly informative Normal prior to the log$_{10}$ titers $v_i$ ($v_i$ is the titer for sample $i$ measured in log$_{10}$ TCID$_{50}$/[0.1 mL], since wells were inoculated with 0.1 mL), similar to that used in our previous work (**Fischer et al., 2020**):

$$v_i \sim \text{Normal}(2.5, 4) \tag{39}$$

## Titer inference model predictive checks

We assessed the appropriateness of this prior distribution choice using prior predictive checks. The prior checks suggested that prior distributions were agnostic over the titer values of interest (**Appendix 1—figure 2**, **Appendix 1—figure 3**).

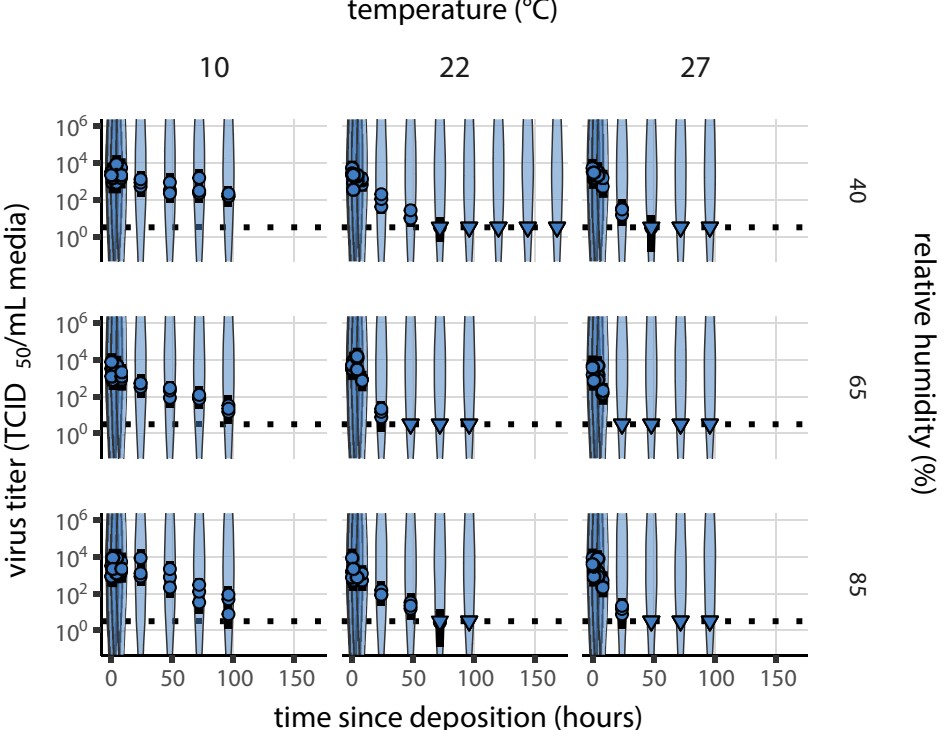

**Appendix 1—figure 2.** Prior predictive check for SARS-CoV-2 titer inference. Violin plots show distribution of simulated titers sampled from the prior predictive distribution. Points show posterior median estimated titers in $\log_{10}$ $TCID_{50}$/mL for each sample; lines show 95% credible intervals. Time-points with no positive wells for any replicate are plotted as triangles at the approximate single-replicate limit of detection (LOD) of the assay—denoted by a black dotted line at $10^{0.5}$ $TCID_{50}$/mL media—to indicate that a range of sub-LOD values are plausible. Three samples collected at each time-point. x-axis shows time since sample deposition. Wide coverage of violins relative to datapoints shows that priors are agnostic over the titer values of interest.

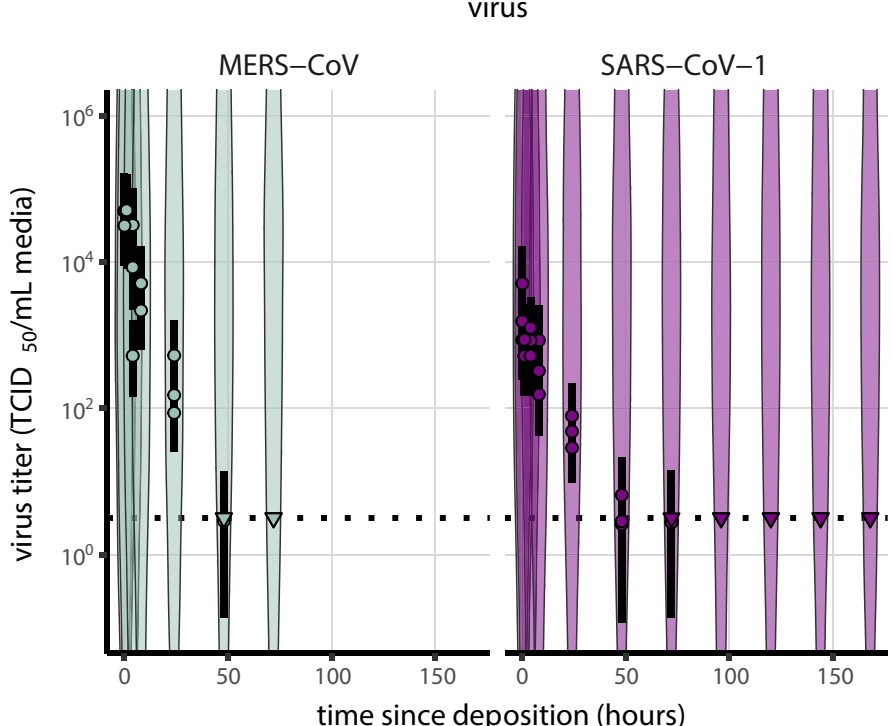

**Appendix 1—figure 3.** Prior predictive check for titer inference for SARS-CoV-1 and MERS-CoV. Violin plots show distribution of simulated titers sampled from the prior predictive distribution. Points show posterior median estimated titers in $\log_{10}$ TCID$_{50}$/mL for each sample; lines show 95% credible intervals. Time-points with no positive wells for any replicate are plotted as triangles at the approximate single-replicate limit of detection (LOD) of the assay—denoted by a black dotted line at $10^{0.5}$ TCID$_{50}$/mL media—to indicate that a range of sub-LOD values are plausible. Three samples collected at each time-point. x-axis shows time since sample deposition. Wide coverage of violins relative to datapoints shows that priors are agnostic over the titer values of interest.

## Evaporation model fitting

Following Modeling of medium evaporation, *Equation 13*, we modeled the expected mass $\bar{m}_i(t)$ for each evaporation experiment $i$ according to the equation:

$$\bar{m}_i(t) = \begin{cases} m_i(0) - \beta_i t & \beta_i t < m_i(0) - m_i(\infty) \\ m_i(\infty) & \text{otherwise} \end{cases} \tag{40}$$

We modeled that the observed masses $m_i(t)$ as normally distributed about the predicted masses $\bar{m}_i(t)$ with an estimated, experiment-specific standard deviation $\sigma_{ei}$:

$$m_i(t) \sim \text{Normal}(\bar{m}_i(t), \sigma_{ei}) \tag{41}$$

To make evaporation prior distributions more interpretable, we placed our prior not on the evaporative mass loss rate $\beta_i$ but rather on the time to reach quasi-equilibrium $\tau_i$ which is related to $\beta_i$ by *Equation 17*:

$$\tau_i = \frac{m_i(0) - m_i(\infty)}{\beta_i}$$

We placed weakly informative Half-Normal priors on the times to quasi-equilibrium $\tau_i$ (measured in hours) and on the measurement standard deviations $\sigma_{ei}$:

$$\tau_i \sim \mathrm{HalfNormal}(10, 10) \tag{42}$$

$$\sigma_{ei} \sim \mathrm{HalfNormal}(0, 1) \tag{43}$$

## Empirical virus decay estimation
### Simple regression model

The duration of virus detectability depends not only on environmental conditions and treatment method but also initial inoculum and sampling noise. We therefore estimated the exponential decay rates of viable virus (and thus virus half-lives) using a simple Bayesian regression approach analogous to that described in *Fischer et al., 2020*. This modeling approach allowed us to account for differences in initial inoculum levels across samples as well as other sources of experimental noise. The model yields estimates of posterior distributions of viral decay rates and half-lives in the various experimental conditions—that is, estimates of the range of plausible values for these parameters given our data, with an estimate of the overall uncertainty (*Gelman et al., 1995*).

Our data consist of nine different experimental conditions corresponding to the combinations of three temperatures (10°C, 22°C, and 27°C) and three relative humidity levels (40%, 65%, and 85%). For each treatment, three samples were collected at 0, 1, 4, 8, 24, 72, and 96 hr after deposition. We also used this model for our group's SARS-CoV-1 and MERS-CoV data (in the meta-analysis), which had one experimental condition each: 22°C and 40% RH, observed over multiple timepoints. We accounted for evaporation with the same 22°C, 40% RH suspension medium evaporation data used for SARS-CoV-2 at that temperature and humidity (as all the virus inactivation experiments were conducted using the same medium).

We modeled each sample $j$ for experimental condition $i$ as starting with some true initial $\log_{10}$ titer $v_{ij0}$. At the time $t_{ij}$ that it is sampled, it has titer $v_{ij}$. As described above (Evaporation and quasi-equilibrium phases), we partitioned each experiment $i$ into a evaporation phase and a quasi-equilibrium phase according to an estimated quasi-equilibration time $\tau_i$.

We modeled loss of viable virus at quasi-equilibrium as exponential decay at an experiment-specific rate $\lambda_i$. To avoid making assumptions about the correctness of our evaporation phase inactivation model (see Modeling of virus decay dynamics during the evaporation phase), we approximated loss of viable virus during the evaporation phase as exponential decay with one decay rate for each temperature condition (which applies to all associated humidity conditions). That is, the evaporation phase decay rate for experiment $i$ is $l_{T(i)}$, where $T(i)$ denotes the temperature for experiment $i$.

It follows that the quantity $v_{ij}$ of virus sampled at time $t_{ij}$ is given by:

$$v_{ij} = \begin{cases} v_{ij0} - l_{T(i)} t_{ij} & t_{ij} \leq \tau_i \\ v_{ij0} - l_{T(i)} \tau_i - \lambda_i(t_{ij} - \tau_i) & t_{ij} > \tau_i \end{cases} \tag{44}$$

We used the direct-from-well data likelihood function described above, except that instead of estimating individual titers independently, we estimated $\lambda_i$ and $l_{T(i)}$ under the assumption that our observed well data $y_{idk}$ reflected the corresponding predicted titers $v_{ij}$.

To check the robustness of our results to our assumptions about the evaporation phase, we also fit a model only to the quasi-equilibrium phase data, with time measured since quasi-equilibrium was reached. In that model, the intercepts $v_{ij0}$ thus reflect the estimated titer at the time quasi-equilibrium was reached:

$$v_{ij0} - \lambda_i(t_{ij} - \tau_i) \tag{45}$$

We modeled each experiment $i$ as having a mean initial $\log_{10}$ titer $\bar{v}_{i0}$. We modeled the individual $v_{ij0}$ as normally distributed about $\bar{v}_{i0}$ with an estimated, experiment-specific standard deviation $\sigma_i$:

$$v_{ij0} \sim \mathrm{Normal}(\bar{v}_{i0}, \sigma_i) \tag{46}$$

## Simple regression model prior distributions

We placed a Normal prior on the mean initial $\log_{10}$ titers $\bar{v}_{i0}$ to reflect the known inocula, similar to.

$$\bar{v}_{i0} \sim \text{Normal}(2.5, 1) \tag{47}$$

We placed a Half-Normal prior on the standard deviations $\sigma_i$:

$$\sigma_i \sim \text{HalfNormal}(0, 0.5) \tag{48}$$

The allows either for large variation (1 log) about the experiment mean or for substantially less variation, depending on the data. It is similar—though slightly more diffuse—to a prior we used in previous work (*Gamble et al., 2021*).

To encode prior information about the decay rates in an interpretable way, we placed Normal priors on the log half-lives $\ln(\eta_i)$, where $\eta_i = \frac{\log_{10}(2)}{\lambda_i}$ and $\ln(\theta_{T(i)})$, where $\theta_{T(i)} = \frac{\log_{10}(2)}{l_{T(i)}}$. We made the priors weakly informative (diffuse over the biologically plausible half-lives); we verified this with prior predictive checks.

$$\begin{aligned} \ln(\eta_i) &\sim \text{Normal}(\ln(6), 2) \\ \ln(\theta_{T(i)}) &\sim \text{Normal}(\ln(24), 1.25) \end{aligned} \tag{49}$$

We used a larger prior mean for the evaporation phase decay rate based on observations of slow decay of SARS-CoV-2 at moderate temperatures in bulk medium (*Chin et al., 2020*) and similar results for other viruses (*Marr et al., 2019*).

## Simple regression model predictive checks

We assessed the appropriateness of prior distribution choices using prior predictive checks and assessed goodness of fit for the estimated model using posterior predictive checks. Prior checks suggested that prior distributions were agnostic over the parameter values of interest, and posterior checks suggested a good fit of the model to the data. The resultant checks are shown below (*Appendix 1—figures 4–11*).

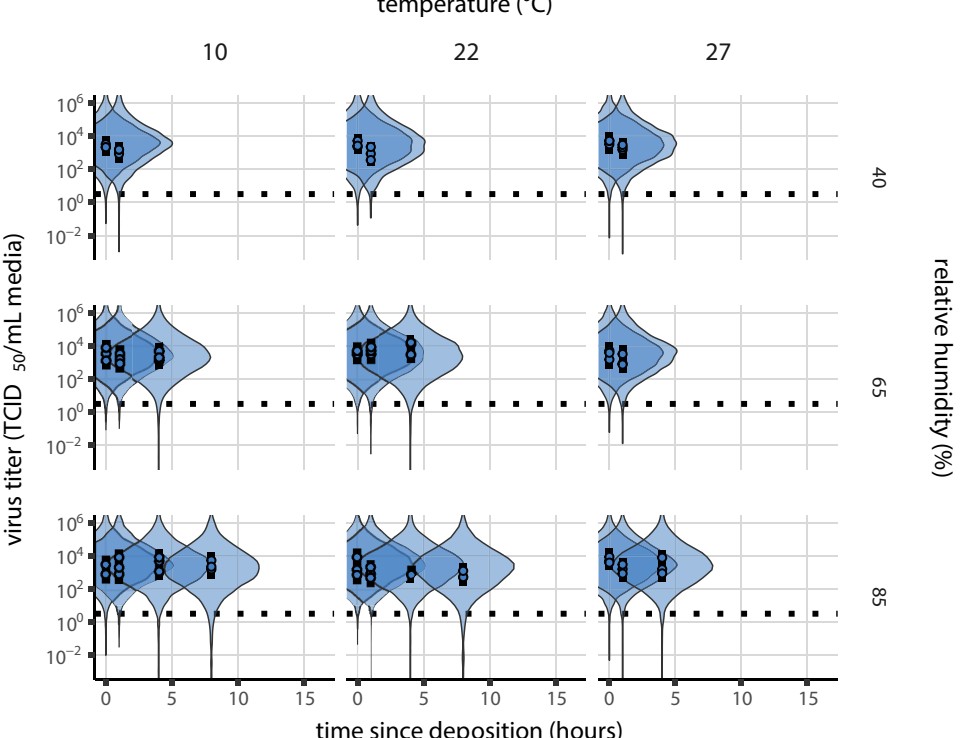

**Appendix 1—figure 4.** Prior predictive check for empirical virus decay during the evaporation phase for SARS-CoV-2. Violin plots show distribution of simulated titers sampled from the prior predictive distribution. Points show posterior median estimated titers in $\log_{10}$ TCID$_{50}$/mL for each sample; lines show 95% credible intervals. x-axis shows time since sample deposition. Black dotted line shows the single-replicate limit of detection of the assay: $10^{0.5}$ TCID$_{50}$/mL media. Wide coverage of violins relative to datapoints show that priors are agnostic over the titer values of interest, and that the priors regard both fast and slow decay rates as possible.

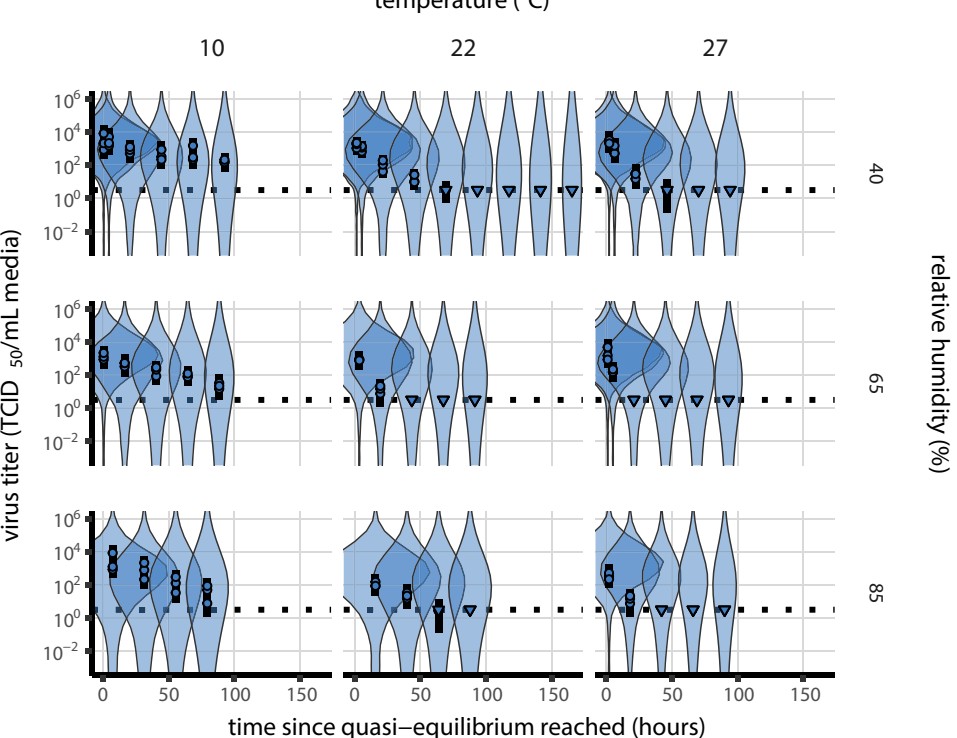

**Appendix 1—figure 5.** Prior predictive check for empirical virus decay at quasi-equilibrium for SARS-CoV-2. Violin plots show distribution of simulated titers sampled from the prior predictive distribution. Points show posterior median estimated titers in $\log_{10}$ TCID$_{50}$/mL for each sample; lines show 95% credible intervals. Time-points with no positive wells for any replicate are plotted as triangles at the approximate single-replicate limit of detection (LOD) of the assay—denoted by a black dotted line at $10^{0.5}$ TCID$_{50}$/mL media—to indicate that a range of sub-LOD values are plausible. Three samples collected at each time-point. x-axis shows time since quasi-equilibrium was reached, as measured in evaporation experiments. Wide coverage of violins relative to datapoints shows that priors are agnostic over the titer values of interest, and that the priors regard both fast and slow decay rates as possible.

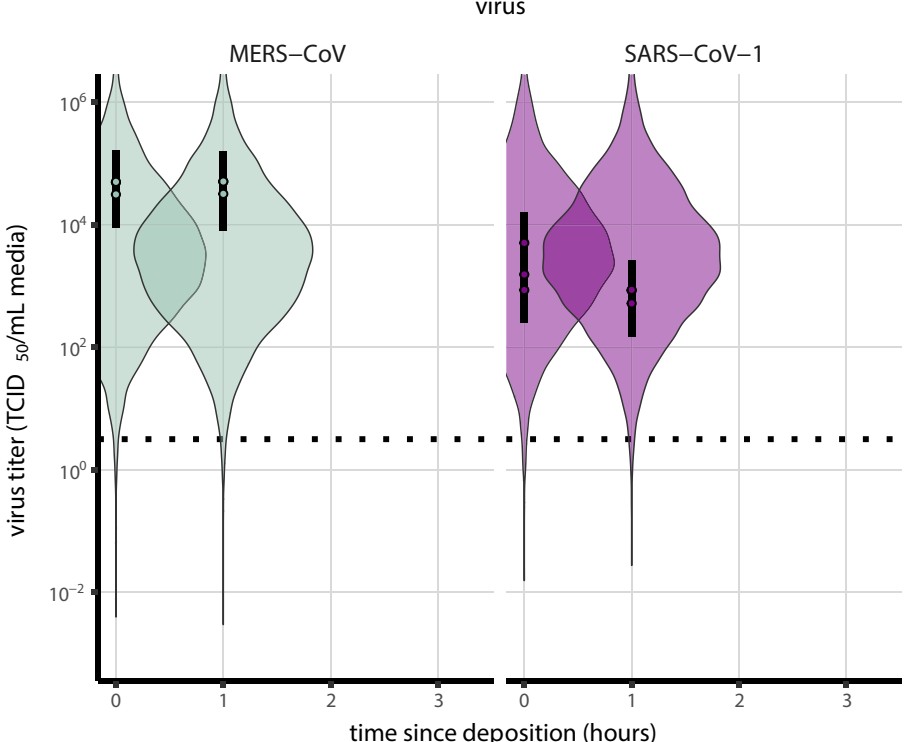

**Appendix 1—figure 6.** Prior predictive check for empirical virus decay during the evaporation phase for SARS-CoV-1 and MERS-CoV at 22°C and 40% relative humidity. Violin plots show distribution of simulated titers sampled from the prior predictive distribution. Points show posterior median estimated titers in $\log_{10}$ $TCID_{50}$/mL for each sample; lines show 95% credible intervals. Black dotted line shows the approximate single-replicate limit of detection (LOD) of the assay: $10^{0.5}$ $TCID_{50}$/mL media. Three samples collected at each time-point. x-axis shows time since sample deposition. Wide coverage of violins relative to datapoints shows that priors are agnostic over the titer values of interest, and that the priors regard both fast and slow decay rates as possible.

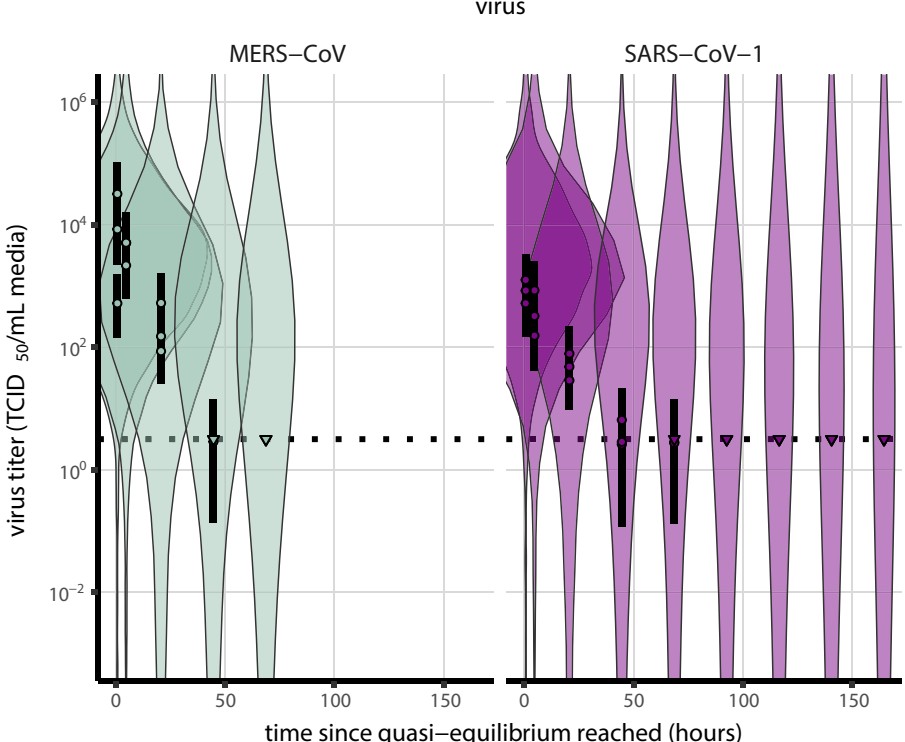

**Appendix 1—figure 7.** Prior predictive check for empirical virus decay at quasi-equilibrium for SARS-CoV-1 and MERS-CoV at 22°C and 40% relative humidity. Violin plots show distribution of simulated titers sampled from the prior predictive distribution. Points show posterior median estimated titers in $\log_{10}$ $TCID_{50}$/mL for each sample; lines show 95% credible intervals. Time-points with no positive wells for any replicate are plotted as triangles at the approximate single-replicate limit of detection (LOD) of the assay—denoted by a black dotted line at $10^{0.5}$ $TCID_{50}$/mL media—to indicate that a range of sub-LOD values are plausible. Three samples collected at each time-point. x-axis shows time since quasi-equilibrium was reached, as measured in evaporation experiments. Wide coverage of violins relative to datapoints shows that priors are agnostic over the titer values of interest, and that the priors regard both fast and slow decay rates as possible.

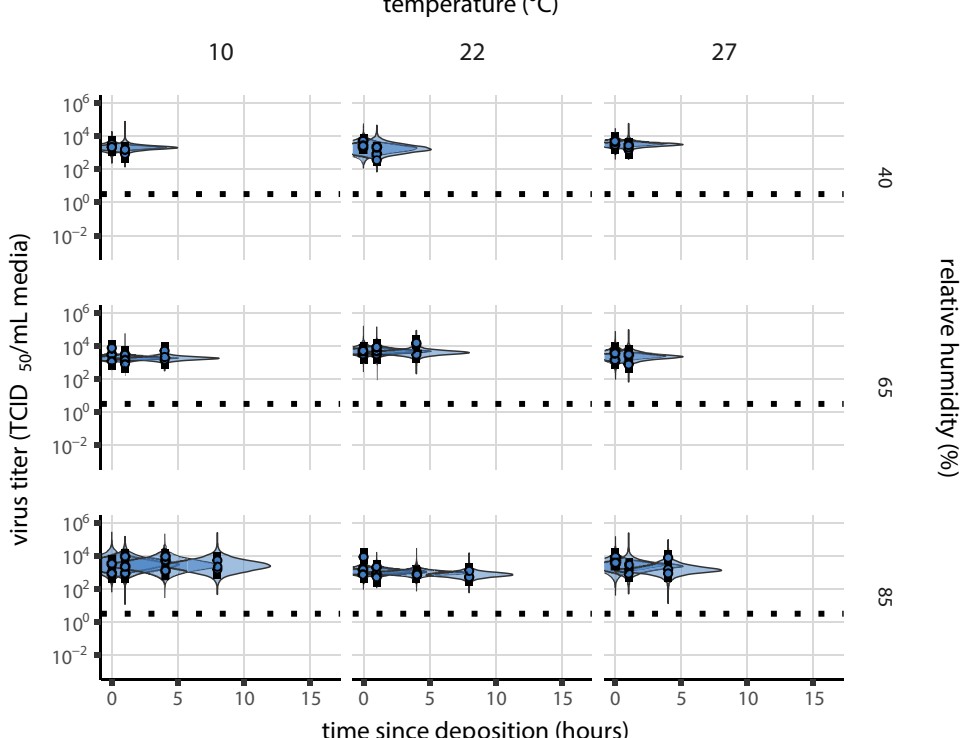

**Appendix 1—figure 8.** Posterior predictive check for empirical virus decay during the evaporation phase for SARS-CoV-2. Violin plots show distribution of simulated titers sampled from the posterior predictive distribution. Points show posterior median estimated titers in $\log_{10}$ TCID$_{50}$/mL for each sample; lines show 95% credible intervals. Black dotted line shows the approximate single-replicate limit of detection (LOD) of the assay: $10^{0.5}$ TCID$_{50}$/mL media. Three samples collected at each time-point. x-axis shows time since sample deposition. Tight correspondence between distribution of posterior simulated titers and independently estimated titers suggests the model fits the data well.

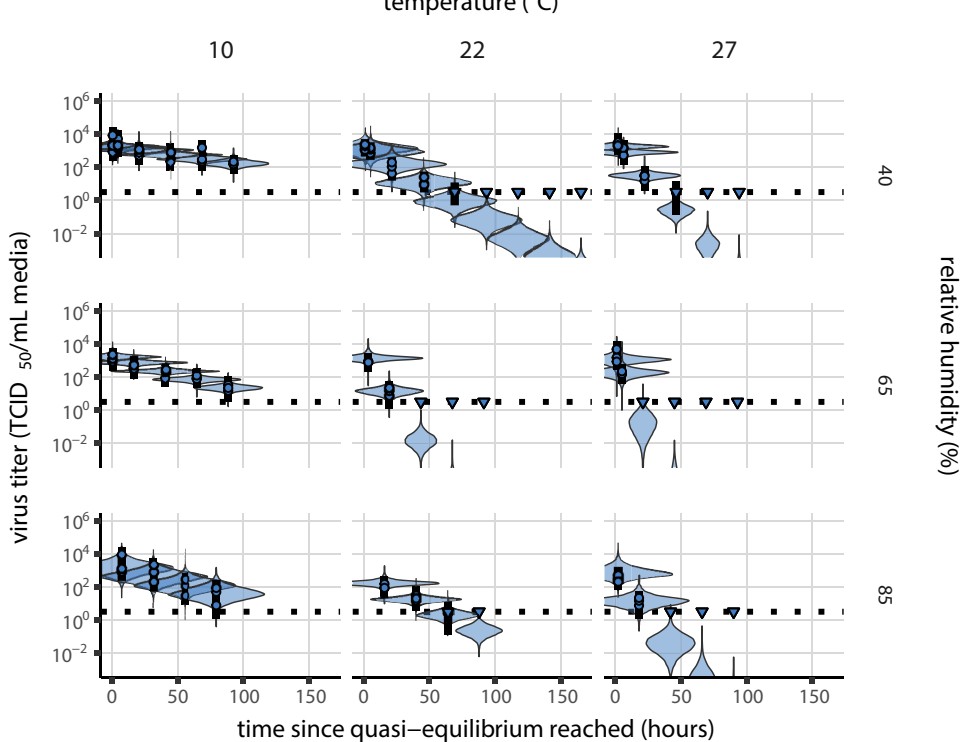

**Appendix 1—figure 9.** Posterior predictive check for empirical virus decay at quasi-equilibrium for SARS-CoV-2. Violin plots show distribution of simulated titers sampled from the posterior predictive distribution. Points show posterior median estimated titers in $\log_{10}$ TCID$_{50}$/mL for each sample; lines show 95% credible intervals. Time-points with no positive wells for any replicate are plotted as triangles at the approximate single-replicate limit of detection (LOD) of the assay—denoted by a black dotted line at $10^{0.5}$ TCID$_{50}$/mL media—to indicate that a range of sub-LOD values are plausible. Three samples collected at each time-point. x-axis shows time since quasi-equilibrium was reached, as measured in evaporation experiments. Tight correspondence between distribution of posterior simulated titers and independently estimated titers suggests the model fits the data well.

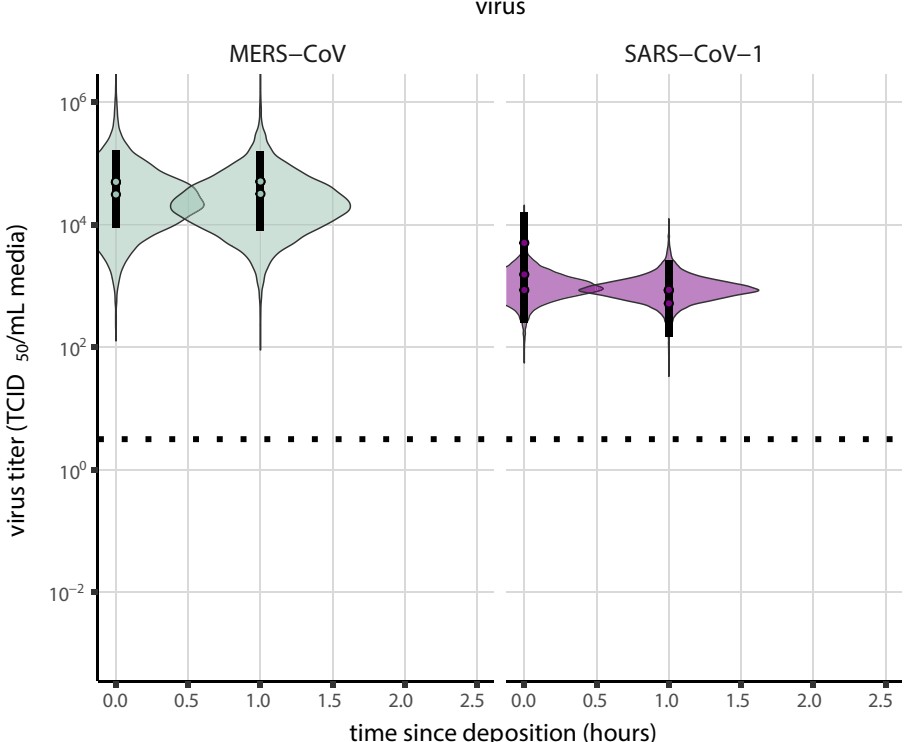

**Appendix 1—figure 10.** Posterior predictive check for empirical virus decay during the evaporation phase for SARS-CoV-1 and MERS-CoV at 22°C and 40% relative humidity. Violin plots show distribution of simulated titers sampled from the posterior predictive distribution. Points show posterior median estimated titers in $\log_{10}$ TCID$_{50}$/mL for each sample; lines show 95% credible intervals. Black dotted line shows the approximate single-replicate limit of detection (LOD) of the assay: $10^{0.5}$ TCID$_{50}$/mL media. Three samples collected at each time-point. x-axis shows time since sample deposition. Tight correspondence between distribution of posterior simulated titers and independently estimated titers suggests the model fits the data well.

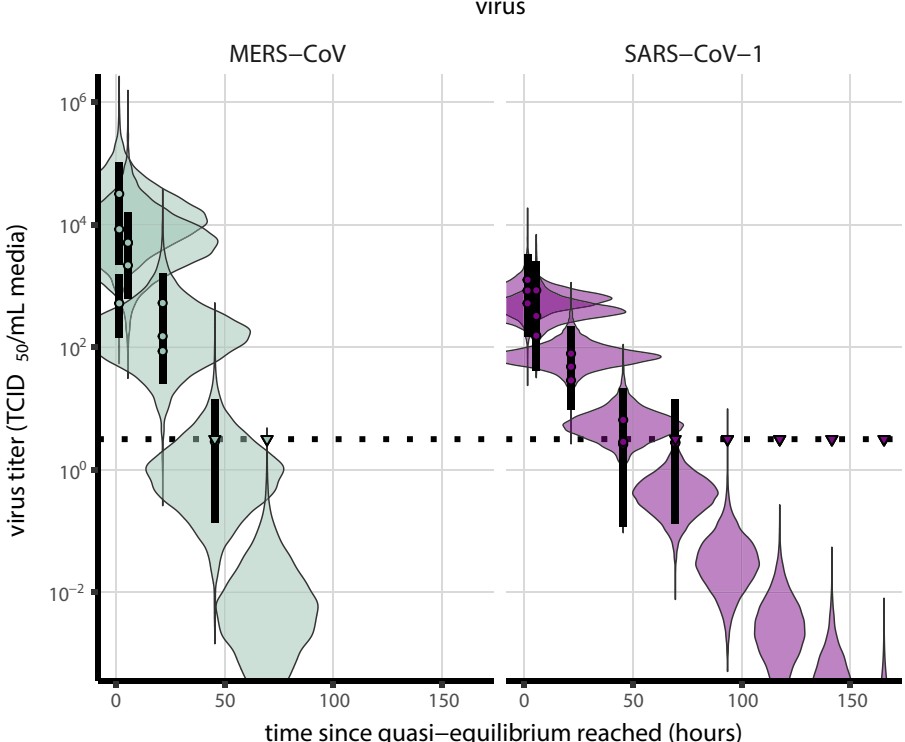

**Appendix 1—figure 11.** Posterior predictive check for empirical virus decay at quasi-equilibrium for SARS-CoV-1 and MERS-CoV at 22°C and 40% relative humidity. Violin plots show distribution of simulated titers sampled from the posterior predictive distribution. Points show posterior median estimated titers in $\log_{10}$ TCID$_{50}$/mL for each sample; lines show 95% credible intervals. Time-points with no positive wells for any replicate are plotted as triangles at the approximate single-replicate limit of detection (LOD) of the assay—denoted by a black dotted line at $10^{0.5}$ TCID$_{50}$/mL media—to indicate that a range of sub-LOD values are plausible. Three samples collected at each time-point. x-axis shows time since quasi-equilibrium was reached, as measured in evaporation experiments. Tight correspondence between distribution of posterior simulated titers and independently estimated titers suggests the model fits the data well.

## Mechanistic model estimation
### Mechanistic model fitting

To fit our mechanistic model (see Mechanistic model for temperature and humidity effects), we partitioned experiments according to humidity into two groups: sub-ERH/efflorescence (40%) and super-ERH/solution (65%, 85%). As before, we partitioned each experiment into a evaporation phase and a quasi-equilibrium phase (see Evaporation and quasi-equilibrium phases).

As before, we modeled titers $v_{ij}$ by assuming an initial value $v_{ij0}$ and then modeling decay from that value. We modeled decay during the evaporation phase according to *Equation 23* and decay during the quasi-equilibrium phase as exponential at a fixed rate $k_i$.

These rates were functions of the temperatures $T_i$ and quasi-equilibrium concentration factors $\frac{[S_{eq}]}{[S_0]}_i$ according to the mechanistic model.

For all experiments $i$, we modeled decay in solution during the evaporation phase as following *Equation 23*, which follows from the time-varying inactivation rate $k_{ev}(t)$ given in *Equation 19*:

$$k_{\text{ev}}{}^i(t) = \frac{w_i(0)}{w_i(t)} A_{\text{sol}} \exp\left(-\frac{E_a}{RT_i}\right) \tag{50}$$

The use of $A_{\text{sol}}$ reflects the assumption that the virus is in solution during the evaporation phase. The $w$ terms model the dynamic concentration factor.

For the quasi-equilibrium phase, we modeled virus decay as exponential at rate $k_{\text{eff}}$ (**Equation 2**) for efflorescent experiments (at 40% relative humidity) and as exponential at rate $k_{\text{sol}}$ (**Equation 3**) for solution experiments (at 65% or 85% relative humidity).

That is:

$$k_i = \begin{cases} A_{\text{eff}} \exp\left(-\frac{E_a}{RT_i}\right) & h_i < \text{ERH} \\[2ex] \frac{[S_{\text{eq}}]}{[S_0]_i} A_{\text{sol}} \exp\left(-\frac{E_a}{RT_i}\right) & h_i \geq \text{ERH} \end{cases} \tag{51}$$

The resultant titer prediction equation is:

$$v_{ij} = \begin{cases} v_{ij0} + \frac{k_{0i}}{B_i} \log_{10}(1 - B_i\, t_{ij}) & t_{ij} \leq t_i^{\text{eq}} \\[2ex] v_{ij0} + \frac{k_{0i}}{B_i} \log_{10}(1 - B_i\, \tau_i) - k_i(t_{ij} - t_i^{\text{eq}}) & t_{ij} > t_i^{\text{eq}} \end{cases} \tag{52}$$

where $k_{0i} = k_{\text{ev}}{}^i(0)$ and $t_i^{\text{eq}}$ is the modeled time to quasi-equilibrium ($t_i^{\text{eq}} = \bar{\tau}_i$ for the main model fit and $t_i^{\text{eq}} = \tau_i$ for the model fit using directly-measured concentration; see Evaporation and quasi-equilibrium phases).

As in the simple regression model, we then used the direct-from-well data likelihood function described above under the assumption that our observed well data $y_{idk}$ reflected the titers $v_{ij}$ predicted by the mechanistic model per **Equation 52**.

We estimated the joint posterior for all parameters. That is, activation energies $E_a$ and asymptotic reaction rates $A$ are estimated in light of evaporative mass loss rates $\beta_i$ and resulting times to quasi-equilibrium $t_i^{\text{eq}}$, and vice versa, for maximally informative propagation of uncertainty.

## Concentration factor

In our evaporation experiments, we measured $m_i(0)$ and $m_i(\infty)$, the initial and final total masses, respectively, of the deposited droplet under the temperature and humidity conditions of experiment $i$.

For experiment $i$, we denote the initial mass of water by $w_i(0)$, the final mass of water by $w_i(\infty)$ and the mass of solutes, which which we assume is conserved, by $s_i$. Then:

$$\begin{aligned} m_i(0) &= w_i(0) + s_i \\ m_i(\infty) &= w_i(\infty) + s_i \end{aligned} \tag{53}$$

Denote the initial and final mass fractions of solutes in experiment $i$ by $Y_i(0) = \frac{s_i}{w_i(0) + s_i}$ and $Y_i(\infty) = \frac{s_i}{w_i(\infty) + s_i}$, respectively.

We treated the $Y_i(0)$ as an estimated parameter, assuming that it had the same value across all experiments: $Y_i(0) = Y(0)$.

To estimate the parameters $\alpha_c$ and $\alpha_s$ for $\frac{[S_{\text{eq}}]}{[S_0]}$ as a function of $h$, we needed to predict the observed final total mass, $m(\infty)$ as a function of $h$.

By definition:

$$m_i(\infty) = \frac{s_i}{Y_i(\infty)} \tag{54}$$

We can find $Y_i(\infty)$ by using the fact that $\frac{Y(\infty)}{1 - Y(\infty)} = \frac{X(\infty)}{1 - X(\infty)} = r(\infty)$, where $X(\infty)$ is the quasi-equilibrium molar fraction of solutes. So $Y_i(\infty) = \frac{r_i(\infty)}{r_i(\infty) + 1}$. Since $s_i = Y_i(0)m_i(0)$, it follows that the predicted quasi-equilibrium total mass for experiment $i$, $\bar{m}_i(\infty)$, is:

$$\bar{m}_i(\infty) = \frac{r_i(\infty) + 1}{r_i(\infty)} Y_i(0) m_i(0) \tag{55}$$

We modeled $r_i(\infty)$ according to **Equation 11**. Using **Equation 55**, we estimated $Y(0)$ and the parameters $\alpha_c$ and $\alpha_s$ of **Equation 11** from our data. We modeled the observed log final total masses $\ln(m_i(\infty))$ as normally distributed about the log predicted quasi-equilibrium total masses $\ln(\bar{m}_i(\infty))$ with an estimated standard deviation $\sigma_m$:

$$\ln(m_i(\infty)) \sim \mathrm{Normal}(\ln(\bar{m}_i(\infty)), \sigma_m) \tag{56}$$

We assumed that quasi-equilibrium total mass values measured below the ERH were equivalent to the quasi-equilibrium total mass values at the ERH; this allowed us to use the 40% RH (sub-ERH) evaporation data points to add additional resolution to the estimation of $\alpha_c$ and $\alpha_s$.

## Mechanistic model versions

As described in the Main Text, we fit the mechanistic model in two ways. The results plotted in our figures use a semi-mechanistic fitted curve estimating the effect of relative humidity on $\frac{[S_{\mathrm{eq}}]}{[S_0]}$. We jointly estimate the mechanistic parameters and the fitted parameters approximating the relationship between RH and $\frac{[S_{\mathrm{eq}}]}{[S_0]}$ (see Solute concentration factor). This allows us to conduct a more principled extrapolation to unobserved RH values.

To check the robustness of our results, we also fit the mechanistic model using only the directly-measured concentration factors obtained from our evaporation experiments. This fit is the most direct snapshot of the relationship between temperature, concentration factor, and inactivation observed in our data, but it can only predict inactivation rates at RH levels where $\frac{[S_{\mathrm{eq}}]}{[S_0]}$ is known.

## Main model fit

In the main model fit (which uses the fitted curve to relate RH to equilibrium concentration factor), we calculated $\frac{[S_{\mathrm{eq}}]}{[S_0]}_i$ from the ambient relative humidity according to **Equation 12**, substituting $\frac{1-Y(0)}{Y(0)}$ for $\frac{1}{r(0)} = \frac{1-X(0)}{X(0)}$, since the two ratios are equal:

$$\frac{[S_{\mathrm{eq}}]}{[S_0]}_i = \frac{1}{r(0)}\frac{X_i(\infty)}{1-X_i(\infty)} = \left(\frac{1-Y(0)}{Y(0)}\right)\left(\frac{-\ln(h)}{\alpha_s}\right)^{\frac{1}{\alpha_c}} \tag{57}$$

Note that this means that $\alpha_s$ and $\alpha_c$ for the main model fit were estimated not only in light of the measured droplet masses but also in light of the measured virus titers, filtered through the mechanistic model of inactivation.

## Directly-measured concentration model fit

In the model fit using directly-measured concentration, we calculated the concentration factor for the $i^{\mathrm{th}}$ experiment, $\frac{[S_{\mathrm{eq}}]}{[S_0]}_i$ according to **Equation 9** using the measured initial and final total masses $m_i(0)$ and $m_i(\infty)$ and the estimated parameter $Y(0)$:

$$\frac{[S_{\mathrm{eq}}]}{[S_0]}_i = \frac{w_i(0)}{w_i(\infty)} = \frac{m_i(0)-s_i}{m_i(\infty)-s_i} = \frac{m_i(0)-Y(0)m_i(0)}{m_i(\infty)-Y(0)m_i(0)} \tag{58}$$

## Mechanistic model prior distributions
### Activation energies and asymptotic reaction rates

To place priors on $E_a$ and $A$ in an interpretable manner, we placed them not on the parameter pairs themselves but rather on the solution and efflorescent half-lives at 20°C, $\eta_{\mathrm{sol}}(20)$ and $\eta_{\mathrm{eff}}(20)$, and the ratios of virus decay rate at 30°C to the virus decay rate at 20°C, $k_{\mathrm{sol}}(30)/k_{\mathrm{sol}}(20)$ and $k_{\mathrm{eff}}(30)/k_{\mathrm{eff}}(20)$. These quantities fully determine the solution and efflorescence $E_a$ and $A$ values.

Decay rate ratios are related to activation energies by:

$$E_a = \frac{R\ln\left(\frac{k(T_1)}{k(T_2)}\right)}{\frac{1}{T_2} - \frac{1}{T_1}} \tag{59}$$

where the temperatures are given in Kelvin.

The 20°C half-lives $\eta(20)$ in hours imply associated exponential decay rates in $\log_{10}$ TCID$_{50}$/mL per hour: $k(20) = \frac{\log_{10}(2)}{n(20)}$. Given an activation energy $E_a$ and a known decay rate $k(T)$ for a given temperature $T$ in Kelvin, one can calculate the asymptotic rate $A$:

$$\ln(A) = \ln(k(T)) + \frac{E_a}{RT} \tag{60}$$

Note that for $A_{\mathrm{sol}}$, this is the asymptotic rate at the initial concentration (i.e. when $\frac{[S(t)]}{[S_0]} = \frac{[S_0]}{[S_0]} = 1$).

We placed a Normal prior on the log of the half-life at 20°C. Since $\eta_{\mathrm{eff}}(20)$ and $\eta_{\mathrm{sol}}(20)$ are the effloresced quasi-equilibrium and unconcentrated solution half-lives, respectively, we used the same prior as that used for the evaporation phase half-life (Simple regression model prior distributions):

$$\begin{aligned}\ln(\eta_{\mathrm{eff}}(20)) &\sim \mathrm{Normal}(\ln(24), 1.25)\\ \ln(\eta_{\mathrm{sol}}(20)) &\sim \mathrm{Normal}(\ln(24), 1.25)\end{aligned} \tag{61}$$

We placed a Half-Normal prior on the natural log of the decay rate ratios:

$$\ln\left(\frac{k(30)}{k(20)}\right) \sim \mathrm{HalfNormal}(0, 1) \tag{62}$$

Note that this means virus inactivation must become more rapid with temperature, another way in which our model's fitted parameters are not truly free, and thus good fits should not necessarily be expected unless the model describes reality.

For fits with distinct $E_a^{\mathrm{sol}}$ and $E_a^{\mathrm{eff}}$, we used the same $\mathrm{HalfNormal}(0, 1)$ prior for both $\ln\left(\frac{k_{\mathrm{eff}}(30)}{k_{\mathrm{eff}}(20)}\right)$ and $\ln\left(\frac{k_{\mathrm{sol}}(30)}{k_{\mathrm{sol}}(20)}\right)$.

## Titer intercepts

We handled the titer intercepts $v_{ij0}$ for the mechanistic model identically to how they were handled in the simple regression model, with identical priors (see **Equation 46**, **Equation 47** and **Equation 48**). We reproduce those equations here for reference:

$$\begin{aligned}v_{ij0} &\sim \mathrm{Normal}(\bar{v}_{i0}, \sigma_i)\\ \bar{v}_{i0} &\sim \mathrm{Normal}(2.5, 1)\\ \sigma_i &\sim \mathrm{HalfNormal}(0, 0.5)\end{aligned}$$

## Concentration factor

We placed a Normal prior on the log of the initial solute mass fraction $Y(0)$, with a mode given by the approximate solute mass fraction for Dulbecco's Modified Eagle Medium (DMEM) reported by the manufacturer (Sigma Aldrich, reference D6546 [**Sigma Aldrich, 2020**]).

$$\ln(Y(0)) \sim \mathrm{Normal}(\ln(0.011), 0.33) \tag{63}$$

We placed Normal priors on the parameters $c_c$ and $c_s$ that model quasi-equilibrium mole fraction ratio as a function of humidity in **Equation 12**:

$$\begin{aligned}c_c &\sim \mathrm{Normal}(0, 0.33)\\ c_s &\sim \mathrm{Normal}(0, 0.33)\end{aligned}$$

Note that this results in lognormal priors on $\alpha_c$ and $\alpha_s$.

We placed a Normal prior on the standard deviation $\sigma_m$ of the observed log quasi-equilibrium mass about its predicted value.

$$\sigma_m \sim \mathrm{Normal}(0, 1) \tag{64}$$

## Mechanistic model predictive checks

We assessed the appropriateness of prior distribution choices using prior predictive checks and assessed goodness of fit for the estimated model using posterior predictive checks. Prior checks suggested that prior distributions were agnostic over the parameter values of interest, and posterior checks suggested a good fit of the model to the data. The resultant checks for the main and directly-measured concentration versions of the mechanistic model of virus decay are shown below (*Appendix 1—figures 12–19*).

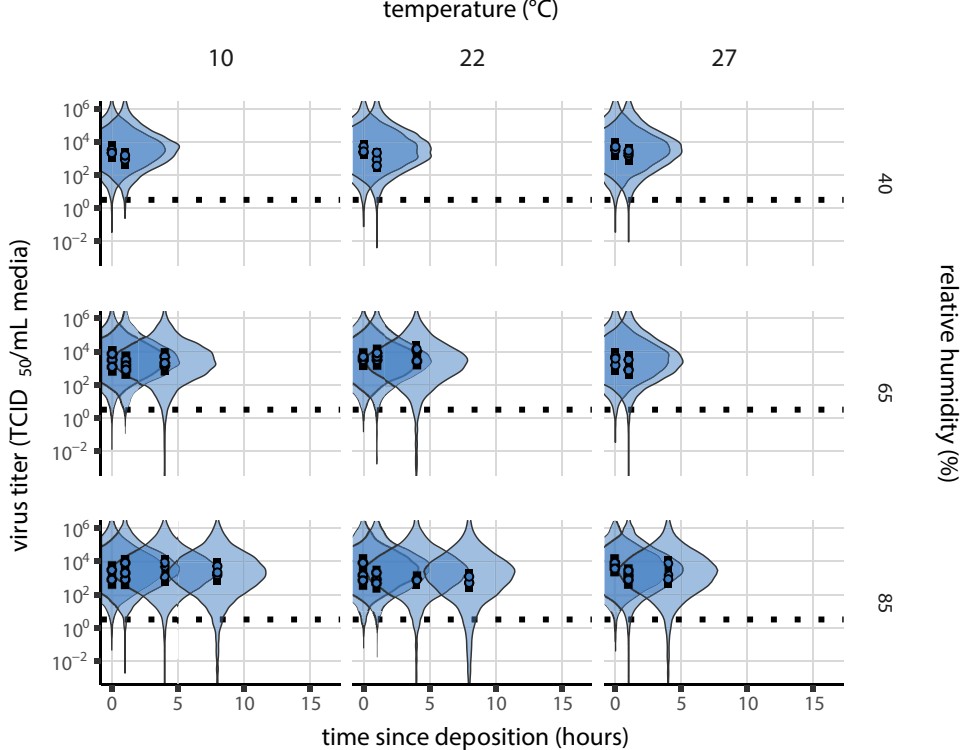

**Appendix 1—figure 12.** Prior predictive check for main model fit during the evaporation phase. Violin plots show distribution of simulated titers sampled from the prior predictive distribution. Points show posterior median estimated titers in $\log_{10}$ TCID$_{50}$/mL for each sample; lines show 95% credible intervals. Black dotted line shows the approximate single-replicate limit of detection (LOD) of the assay: $10^{0.5}$ TCID$_{50}$/mL media. Three samples collected at each time-point. x-axis shows time since sample deposition. Wide coverage of violins relative to datapoints shows that priors are agnostic over the titer values of interest, and that the priors regard both fast and slow decay rates as possible.

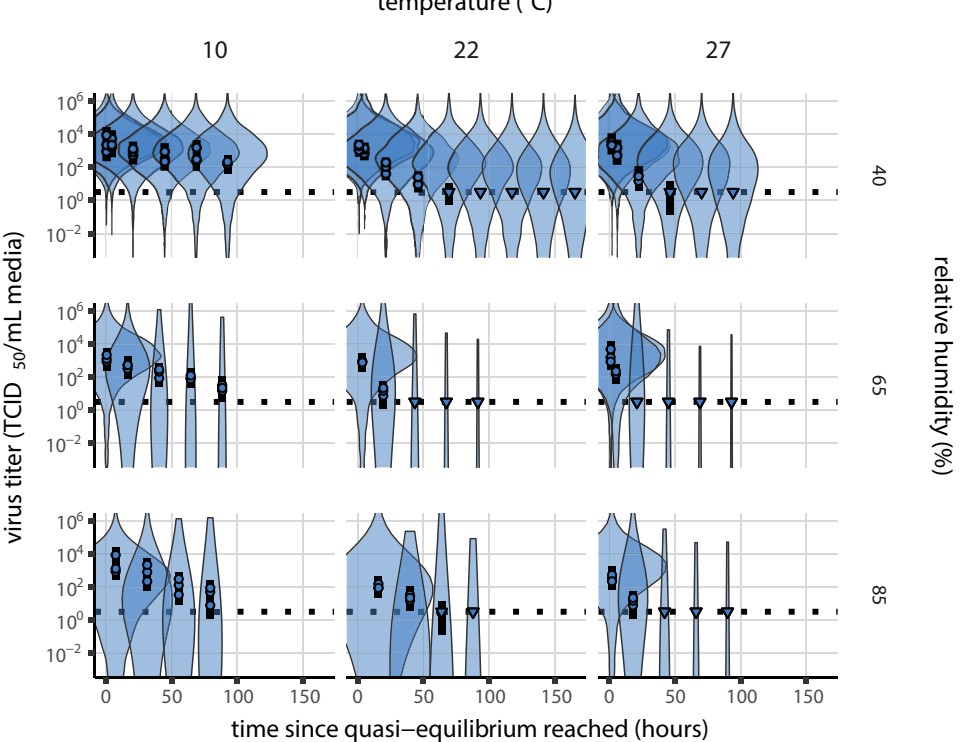

**Appendix 1—figure 13.** Prior predictive check for main model fit at quasi-equilibrium. Violin plots show distribution of simulated titers sampled from the prior predictive distribution. Points show posterior median estimated titers in $\log_{10}$ TCID$_{50}$/mL for each sample; lines show 95% credible intervals. Time-points with no positive wells for any replicate are plotted as triangles at the approximate single-replicate limit of detection (LOD) of the assay—denoted by a black dotted line at $10^{0.5}$ TCID$_{50}$/mL media—to indicate that a range of sub-LOD values are plausible. Three samples collected at each time-point. x-axis shows time since quasi-equilibrium was reached, as measured in evaporation experiments. Wide coverage of violins relative to datapoints shows that priors are agnostic over the titer values of interest, and that the priors regard both fast and slow decay rates as possible.

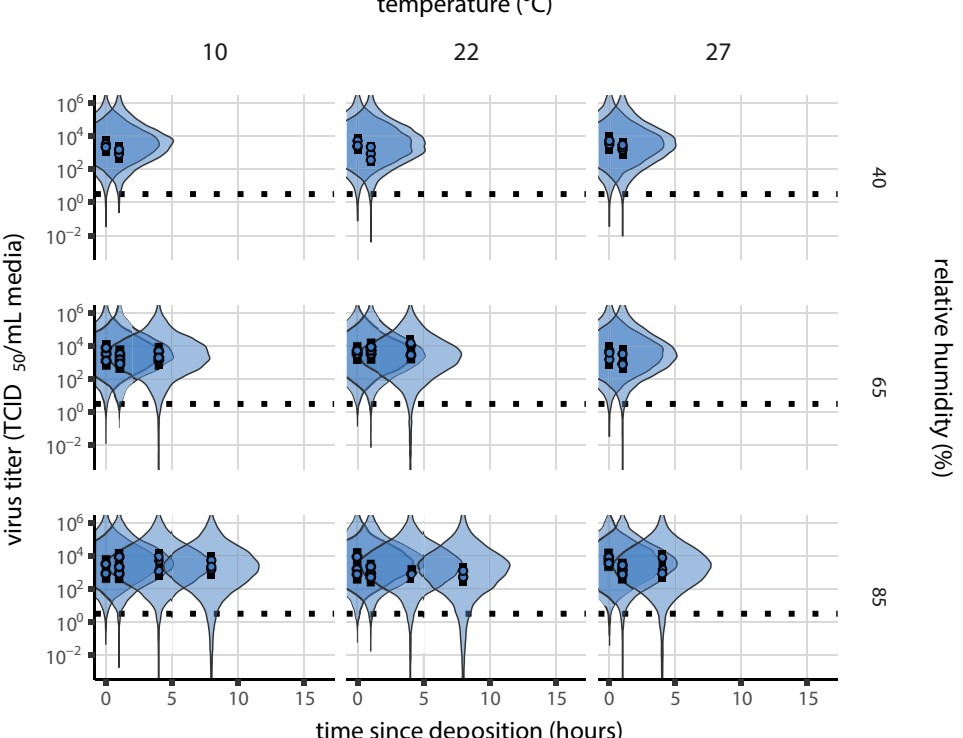

**Appendix 1—figure 14.** Prior predictive check for model fit using directly-measured concentration during the evaporation phase. Violin plots show distribution of simulated titers sampled from the prior predictive distribution. Points show posterior median estimated titers in $\log_{10}$ TCID$_{50}$/mL for each sample; lines show 95% credible intervals. Black dotted line shows the approximate single-replicate limit of detection (LOD) of the assay: $10^{0.5}$ TCID$_{50}$/mL media. Three samples collected at each time-point. x-axis shows time since sample deposition. Wide coverage of violins relative to datapoints shows that priors are agnostic over the titer values of interest, and that the priors regard both fast and slow decay rates as possible.

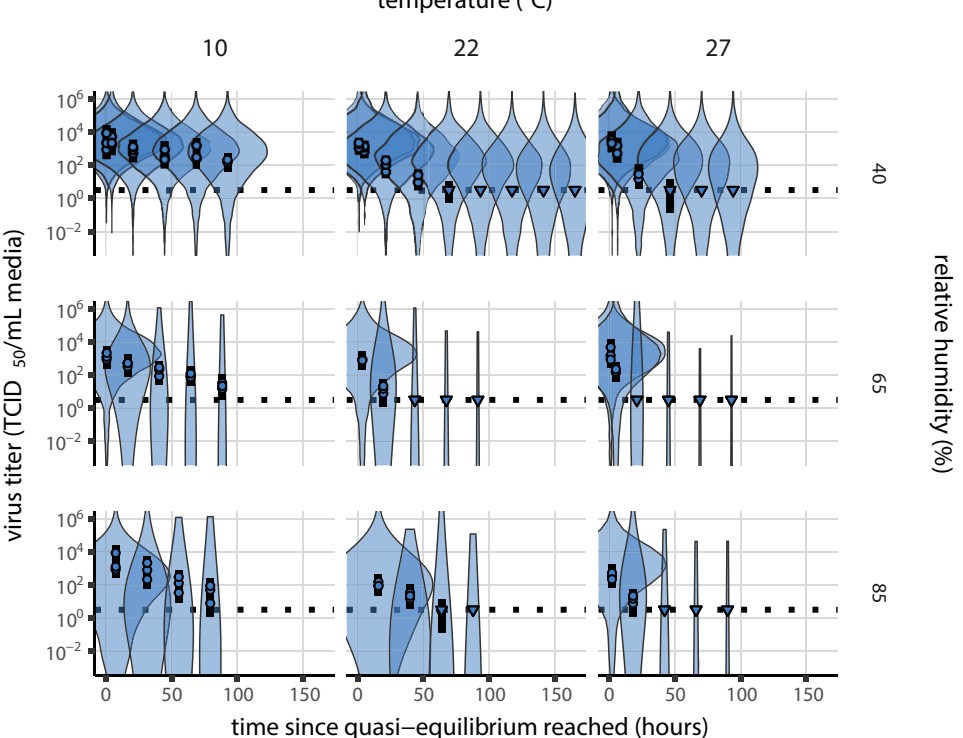

**Appendix 1—figure 15.** Prior predictive check for model fit using directly-measured concentration at quasi-equilibrium. Violin plots show distribution of simulated titers sampled from the prior predictive distribution. Points show posterior median estimated titers in $\log_{10}$ TCID$_{50}$/mL for each sample; lines show 95% credible intervals. Time-points with no positive wells for any replicate are plotted as triangles at the approximate single-replicate limit of detection (LOD) of the assay— denoted by a black dotted line at $10^{0.5}$ TCID$_{50}$/mL media—to indicate that a range of sub-LOD values are plausible. Three samples collected at each time-point. x-axis shows time since quasi-equilibrium was reached, as measured in evaporation experiments. Wide coverage of violins relative to datapoints shows that priors are agnostic over the titer values of interest, and that the priors regard both fast and slow decay rates as possible.

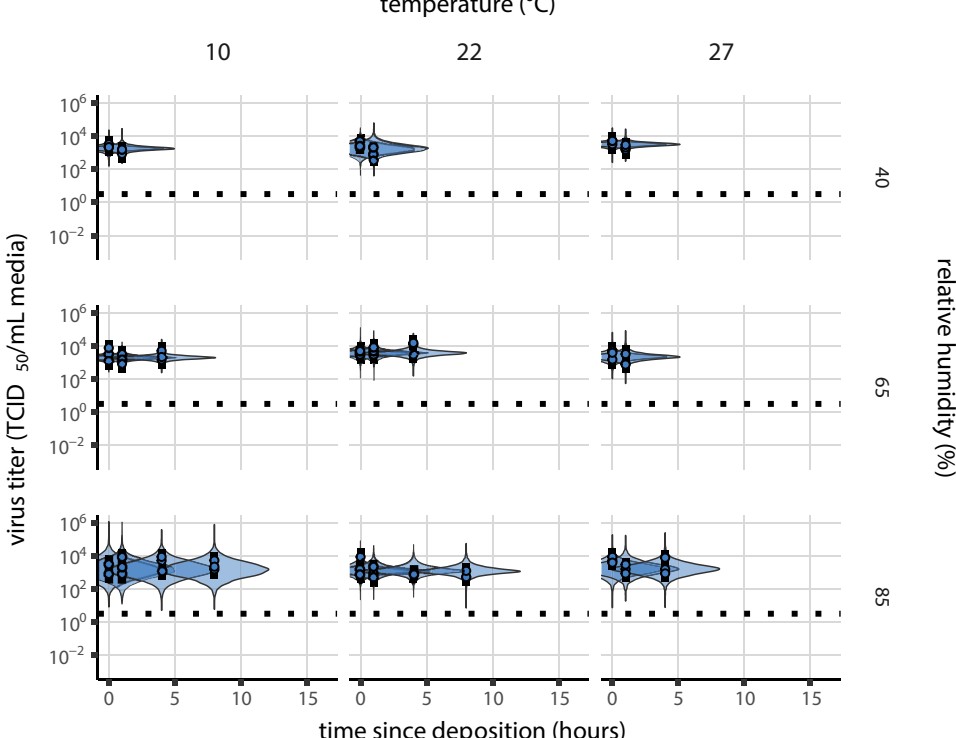

**Appendix 1—figure 16.** Posterior predictive check for main model fit during the evaporation phase. Violin plots show distribution of simulated titers sampled from the posterior predictive distribution. Points show posterior median estimated titers in $\log_{10}$ TCID$_{50}$/mL for each sample; lines show 95% credible intervals. Black dotted line shows the approximate single-replicate limit of detection (LOD) of the assay: $10^{0.5}$ TCID$_{50}$/mL media. Three samples collected at each time-point. x-axis shows time since sample deposition. Tight correspondence between distribution of posterior simulated titers and independently estimated titers suggests the model fits the data well.

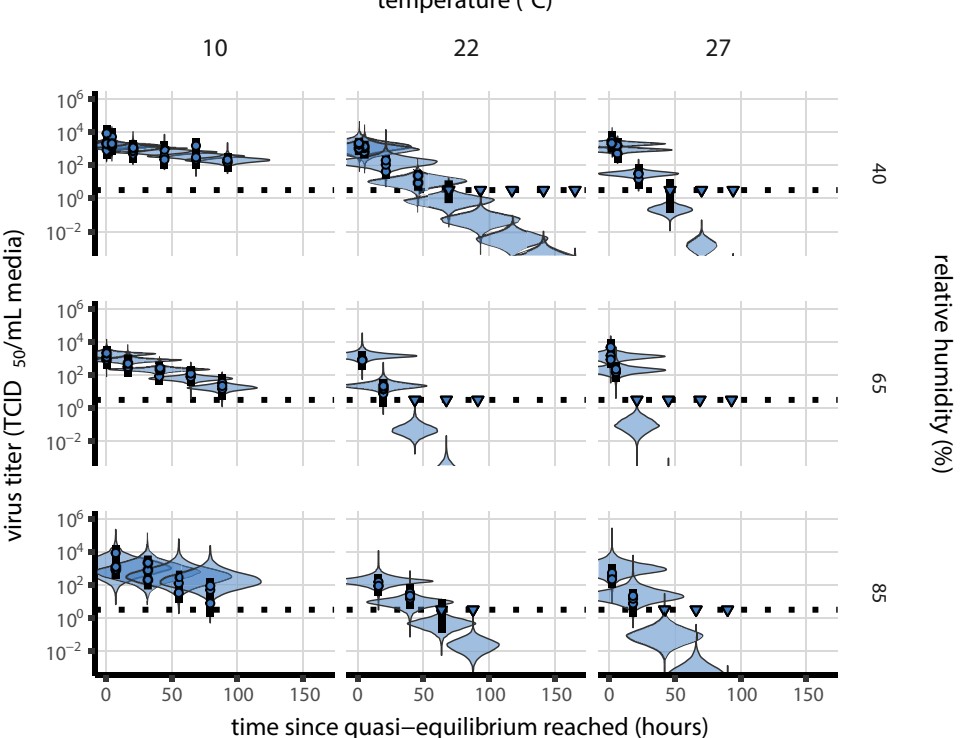

**Appendix 1—figure 17.** Posterior predictive check for main model fit at quasi-equilibrium. Violin plots show distribution of simulated titers sampled from the posterior predictive distribution. Points show posterior median estimated titers in $\log_{10}$ TCID$_{50}$/mL for each sample; lines show 95% credible intervals. Time-points with no positive wells for any replicate are plotted as triangles at the approximate single-replicate limit of detection (LOD) of the assay—denoted by a black dotted line at $10^{0.5}$ TCID$_{50}$/mL media—to indicate that a range of sub-LOD values are plausible. Three samples collected at each time-point. x-axis shows time since quasi-equilibrium was reached, as measured in evaporation experiments. Tight correspondence between distribution of posterior simulated titers and independently estimated titers suggests the model fits the data well.

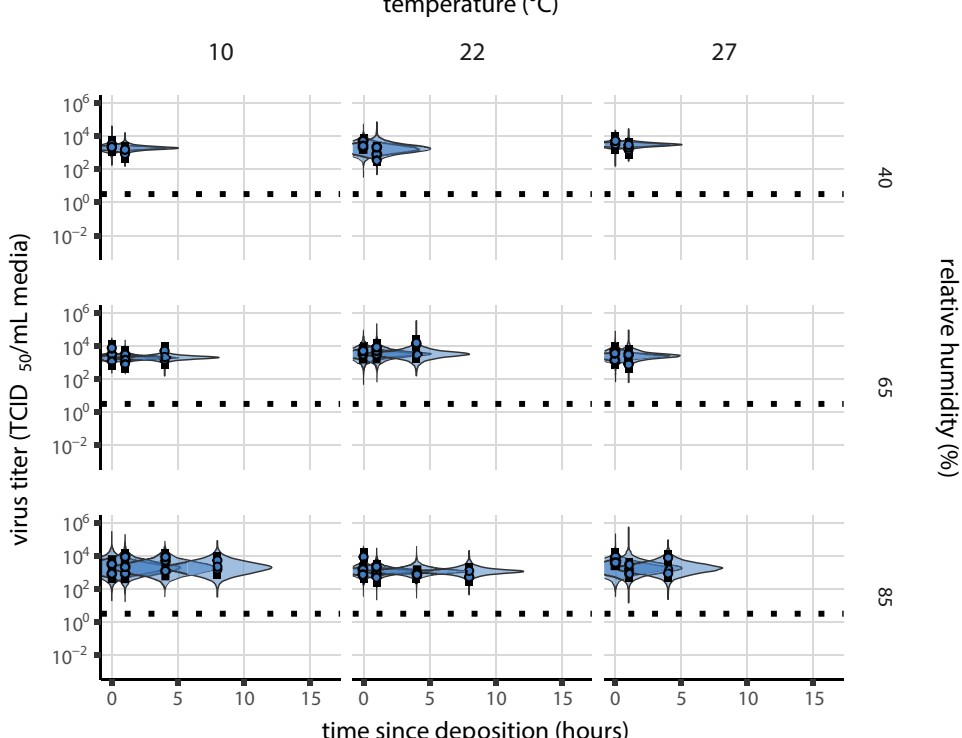

**Appendix 1—figure 18.** Posterior predictive check for model fit using directly-measured concentration during the evaporation phase. Violin plots show distribution of simulated titers sampled from the posterior predictive distribution. Points show posterior median estimated titers in $\log_{10}$ TCID$_{50}$/mL for each sample; lines show 95% credible intervals. Black dotted line shows the approximate single-replicate limit of detection (LOD) of the assay: $10^{0.5}$ TCID$_{50}$/mL media. Three samples collected at each time-point. x-axis shows time since sample deposition. Tight correspondence between distribution of posterior simulated titers and independently estimated titers suggests the model fits the data well.

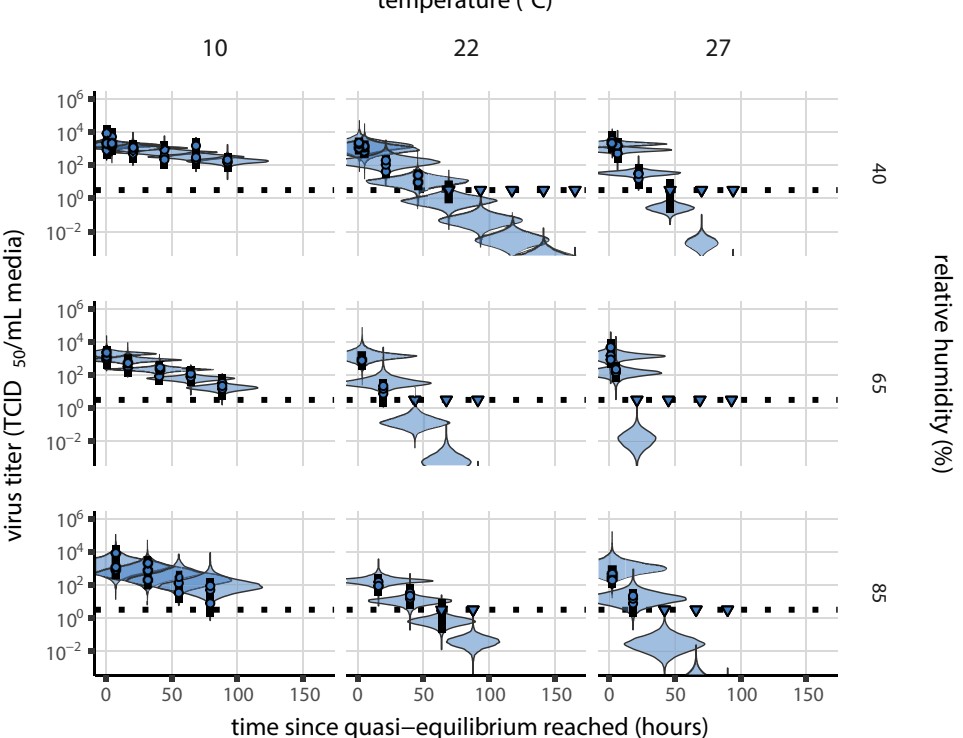

**Appendix 1—figure 19.** Posterior predictive check for model fit using directly-measured concentration at quasi-equilibrium. Violin plots show distribution of simulated titers sampled from the posterior predictive distribution. Points show posterior median estimated titers in $\log_{10}$ TCID$_{50}$/mL for each sample; lines show 95% credible intervals. Time-points with no positive wells for any replicate are plotted as triangles at the approximate single-replicate limit of detection (LOD) of the assay—denoted by a black dotted line at $10^{0.5}$ TCID$_{50}$/mL media—to indicate that a range of sub-LOD values are plausible. Three samples collected at each time-point. x-axis shows time since quasi-equilibrium was reached, as measured in evaporation experiments. Tight correspondence between distribution of posterior simulated titers and independently estimated titers suggests the model fits the data well.

## Meta-analysis of human coronavirus half-lives

### Study selection and data extraction

We screened the Web of Science Core Collection database on May 31, 2020, using the following key words: 'coronavir\* AND (stability OR viability OR inactiv\*) AND (temperature OR heat OR humidity)' (83 records). We also considered opportunistically identified pre-prints (up to July 6, 2020) and studies referenced in full texts assessed for eligibility that reported datasets of potential interest (22 records). We then selected publications reporting virus stability data for human coronaviruses (MERS, SARS-CoV-1, SARS-CoV-2, HCoV-OC43, HCoV-HKU1, HCoV-229E, and HCoV-NL63) with at least two temperature or humidity conditions. Considering the impact of medium composition and contact surface on virus inactivation kinetics (*Yang and Marr, 2012*; *van Doremalen et al., 2020*), we also filtered the selected studies based on these criteria. The complete selection procedure is described in *Appendix 1—figure 20* following the Preferred Reporting Items for Systematic Reviews and Meta-Analyses (PRISMA) (*Moher et al., 2009*). Studies included in our analysis are listed in *Appendix 1—table 3*.

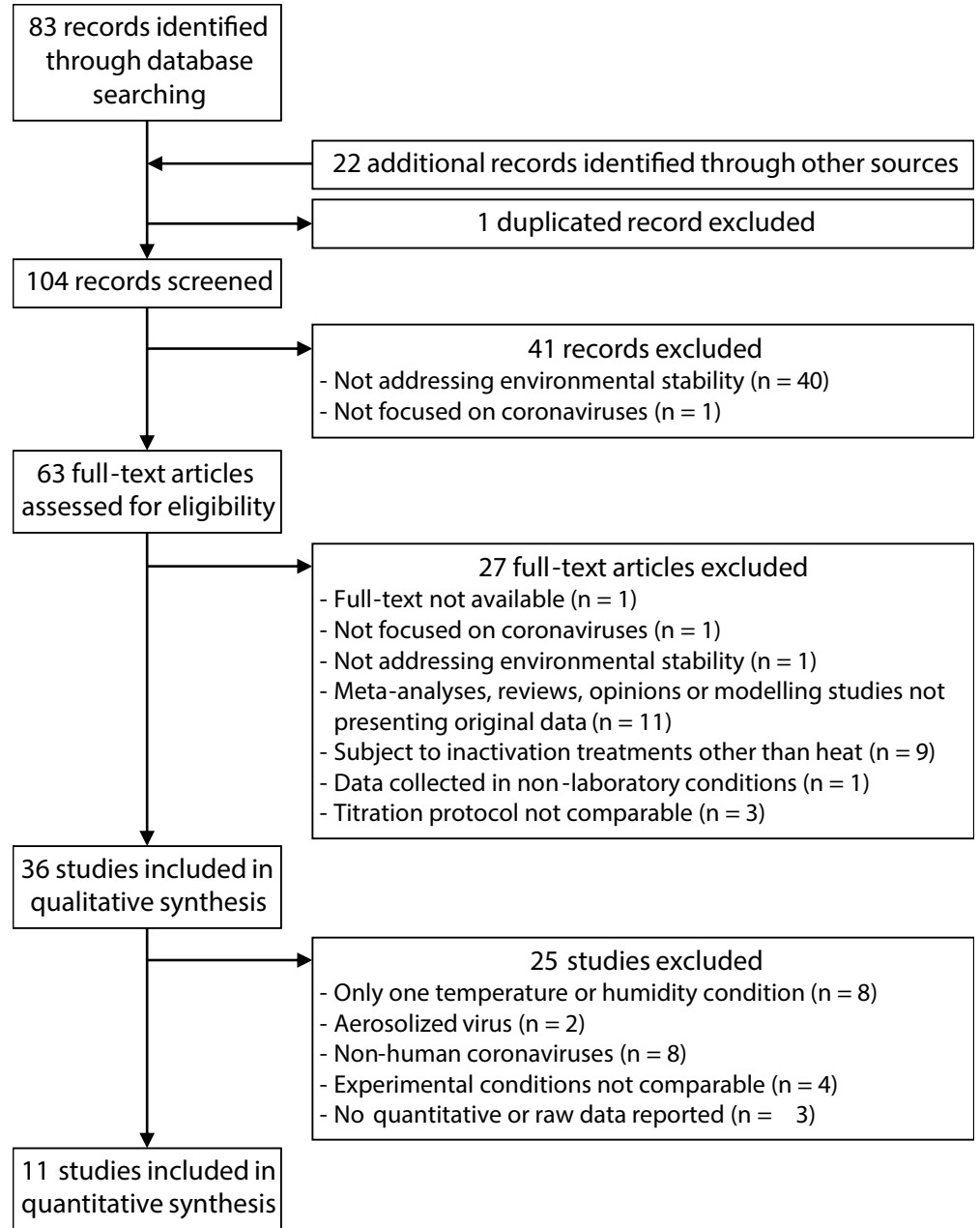

**Appendix 1—figure 20.** Selection process of the studies included in the meta-analysis of the effect of temperature and humidity on human coronaviruses.

We compiled data in the form of viral titer or relative infectivity across time, depending on how they were reported in the selected studies. Data were most often reported as mean ± variation (standard deviation or 95% confidence interval) across replicates per time-point and experimental condition. However, as number of replicates and measured variation was not systematically reported, we did not include this information in our analyses. We extracted data from tables and from figures manually using the WebPlotDigitizer application (*Rohatgi, 2019*). We also recorded metadata including environmental conditions (temperature and relative humidity), contact surface, and medium composition and volume. The complete dataset is available in the online data and code repository.

Among the selected studies, we sub-selected data to be included in our meta-analysis based on the same criteria. In particular, we restricted the dataset to suspensions composed of respiratory secretions, or cell culture or virus transportation media supplemented only with antibiotics and up to 10% fetal calf serum and 1% glutamine; we also restricted the dataset to stability measurements

conducted in bulk medium suspensions, or using droplets deposited on inert surfaces (including steel and polypropylene) or on skin. The final dataset consisted of 38 experimental conditions, covering 17 temperature-humidity combinations and five human coronaviruses (HCoV-229E, HCoV-OC43, MERS-CoV, SARS-CoV-1, and SARS-CoV-2) listed in *Appendix 1—table 3*.

**Appendix 1—table 3.** Estimated half-lives in hours for data from the literature, as a function of material, temperature (T), and relative humidity (RH).
Estimated half-lives are reported as posterior median and the middle 95% credible interval. CCM: cell culture medium; VTM: virus transport medium; Resp. sec.: respiratory secretions.

| Study | Virus | Material | T (°C) | RH (%) | Median half-life (h) | 2.5 % | 97.5 % |
|---|---|---|---|---|---|---|---|
| *Harbourt et al., 2020* | SARS-CoV-2 | Skin | 4 | 45 | $4.18 \times 10^1$ | $2.59 \times 10^1$ | $1.42 \times 10^2$ |
| *Lamarre and Talbot, 1989* | HCoV-229E | Bulk CCM | 4 | | $1.87 \times 10^2$ | $4.96 \times 10^1$ | $8.55 \times 10^3$ |
| *Rabenau et al., 2005* | SARS-CoV-1 | Bulk CCM | 4 | | 1.15 | $1.10 \times 10^{-1}$ | $8.63 \times 10^2$ |
| *Lai et al., 2005* | SARS-CoV-1 | Bulk Resp. sec. | 4 | | $4.42 \times 10^1$ | $3.63 \times 10^1$ | $5.74 \times 10^1$ |
| *Chin et al., 2020* | SARS-CoV-2 | Bulk VTM | 4 | | $1.96 \times 10^2$ | $5.46 \times 10^1$ | $1.06 \times 104$ |
| *van Doremalen et al., 2013* | MERS-CoV | Plastic | 20 | 40 | 1.55 | 1.08 | 2.44 |
| *van Doremalen et al., 2013* | MERS-CoV | Steel | 20 | 40 | 3.16 | 2.29 | 4.77 |
| *Lai et al., 2005* | SARS-CoV-1 | Bulk Resp. sec. | 20 | | $1.10 \times 10^1$ | 8.38 | $1.61 \times 10^1$ |
| *Harbourt et al., 2020* | SARS-CoV-2 | Skin | 22 | 45 | 3.75 | 2.07 | 8.06 |
| *Lamarre and Talbot, 1989* | HCoV-229E | Bulk CCM | 22 | | $1.52 \times 10^1$ | 8.83 | $2.57 \times 10^1$ |
| *Chin et al., 2020* | SARS-CoV-2 | Bulk VTM | 22 | | $1.84 \times 10^1$ | $1.34 \times 10^1$ | $2.64 \times 10^1$ |
| *van Doremalen et al., 2013* | MERS-CoV | Plastic | 30 | 30 | 1.18 | $6.27 \times 10^{-1}$ | 2.34 |
| *van Doremalen et al., 2013* | MERS-CoV | Steel | 30 | 30 | 1.31 | $7.19 \times 10^{-1}$ | 2.60 |
| *van Doremalen et al., 2013* | MERS-CoV | Plastic | 30 | 80 | $9.66 \times 10^{-1}$ | $5.56 \times 10^{-1}$ | 1.78 |
| *van Doremalen et al., 2013* | MERS-CoV | Steel | 30 | 80 | $5.74 \times 10^{-1}$ | $3.99 \times 10^{-1}$ | $9.69 \times 10^{-1}$ |
| *Bucknall et al., 1972* | HCoV-229E | Bulk CCM | 33 | | 1.61 | 1.00 | 3.85 |
| *Bucknall et al., 1972* | HCoV-OC43 | Bulk CCM | 33 | | 6.55 | 3.72 | $7.08 \times 10^1$ |
| *Lamarre and Talbot, 1989* | HCoV-229E | Bulk CCM | 33 | | $1.43 \times 10^1$ | 8.55 | $2.29 \times 10^1$ |
| *Harbourt et al., 2020* | SARS-CoV-2 | Skin | 37 | 45 | $5.96 \times 10^{-1}$ | $3.37 \times 10^{-1}$ | 1.32 |
| *Bucknall et al., 1972* | HCoV-229E | Bulk CCM | 37 | | 1.04 | $6.60 \times 10^{-1}$ | 2.19 |
| *Bucknall et al., 1972* | HCoV-OC43 | Bulk CCM | 37 | | 4.22 | 2.30 | $6.53 \times 10^1$ |
| *Lamarre and Talbot, 1989* | HCoV-229E | Bulk CCM | 37 | | 5.73 | 2.89 | $1.11 \times 10^1$ |

*Continued on next page*

*Appendix 1—table 3 continued*

| Study | Virus | Material | T (°C) | RH (%) | Median half-life (h) | 2.5 % | 97.5 % |
|---|---|---|---|---|---|---|---|
| *Chin et al., 2020* | SARS-CoV-2 | Bulk VTM | 37 | | 2.09 | 1.48 | 3.12 |
| *Batéjat et al., 2021* | SARS-CoV-2 | Bulk CCM | 56 | | $2.25 \times 10^{-2}$ | $1.65 \times 10^{-2}$ | $2.80 \times 10^{-2}$ |
| *Darnell et al., 2004* | SARS-CoV-1 | Bulk CCM | 56 | | $4.49 \times 10^{-2}$ | $3.45 \times 10^{-2}$ | $6.34 \times 10^{-2}$ |
| *Leclercq et al., 2014* | MERS-CoV | Bulk CCM | 56 | | $4.32 \times 10^{-3}$ | $1.27 \times 10^{-3}$ | $1.52 \times 10^{-2}$ |
| *Rabenau et al., 2005* | SARS-CoV-1 | Bulk CCM | 56 | | $3.08 \times 10^{-3}$ | $1.08 \times 10^{-5}$ | $2.63 \times 10^{-2}$ |
| *Chin et al., 2020* | SARS-CoV-2 | Bulk VTM | 56 | | $1.64 \times 10^{-2}$ | $1.07 \times 10^{-2}$ | $2.89 \times 10^{-2}$ |
| *Pagat et al., 2007* | SARS-CoV-1 | Bulk CCM | 60 | | $3.49 \times 10^{-2}$ | $2.61 \times 10^{-2}$ | $5.06 \times 10^{-2}$ |
| *Rabenau et al., 2005* | SARS-CoV-1 | Bulk CCM | 60 | | $3.16 \times 10^{-3}$ | $9.01 \times 10^{-6}$ | $2.63 \times 10^{-2}$ |
| *Batéjat et al., 2021* | SARS-CoV-2 | Bulk CCM | 65 | | $1.86 \times 10^{-3}$ | $7.62 \times 10^{-6}$ | $1.17 \times 10^{-2}$ |
| *Darnell et al., 2004* | SARS-CoV-1 | Bulk CCM | 65 | | $4.14 \times 10^{-2}$ | $3.08 \times 10^{-2}$ | $6.21 \times 10^{-2}$ |
| *Leclercq et al., 2014* | MERS-CoV | Bulk CCM | 65 | | $7.35 \times 10^{-4}$ | $5.15 \times 10^{-4}$ | $1.42 \times 10^{-3}$ |
| *Batéjat et al., 2021* | SARS-CoV-2 | Bulk Resp. sec. | 65 | | $8.36 \times 10^{-3}$ | $5.95 \times 10^{-3}$ | $1.16 \times 10^{-2}$ |
| *Pagat et al., 2007* | SARS-CoV-1 | Bulk CCM | 70 | | $9.79 \times 10^{-3}$ | $7.72 \times 10^{-3}$ | $1.38 \times 10^{-2}$ |
| *Chin et al., 2020* | SARS-CoV-2 | Bulk VTM | 70 | | $3.28 \times 10^{-3}$ | $1.69 \times 10^{-3}$ | $6.10 \times 10^{-3}$ |
| *Darnell et al., 2004* | SARS-CoV-1 | Bulk CCM | 75 | | $2.31 \times 10^{-3}$ | $8.88 \times 10^{-6}$ | $2.10 \times 10^{-2}$ |
| *Batéjat et al., 2021* | SARS-CoV-2 | Bulk Resp. sec. | 95 | | $2.41 \times 10^{-3}$ | $1.62 \times 10^{-3}$ | $3.62 \times 10^{-3}$ |

## Estimation of virus decay in the literature

### Estimation model and priors

We converted all data from the literature into $\log_{10}$ fraction of viable virus remaining (*Appendix 1—figures 21* and *22*). That is, we normalized the reported quantity of viable virus to the earliest measurement—if the authors had not already done so—and expressed time as time elapsed since earliest measurement. We then estimated half-lives independently for each environmental condition $j$ in each study $i$ by fitting a Bayesian exponential decay model with exponential decay rates $\lambda_{ij}$ for each experiment $j$. We treated each reported measurement $y_{ijk}$ (in $\log_{10}$ fraction viable) from experiment $j$ of study $i$ as normally distributed about the predicted $\log_{10}$ fraction viable $\bar{f}_{ijk}$, with an unknown standard deviation $\sigma_{\mathrm{mat}}(i,j)$ estimated independently for each material in study $i$, but shared across all temperature/humidity conditions for that study-material pair.

$$y_{ijk} \sim \mathrm{Normal}\big(\bar{f}_{ijk}, \sigma_{\mathrm{mat}}(i,j)\big)$$
$$\bar{f}_{ijk} = -\lambda_{ij}t$$

(65)

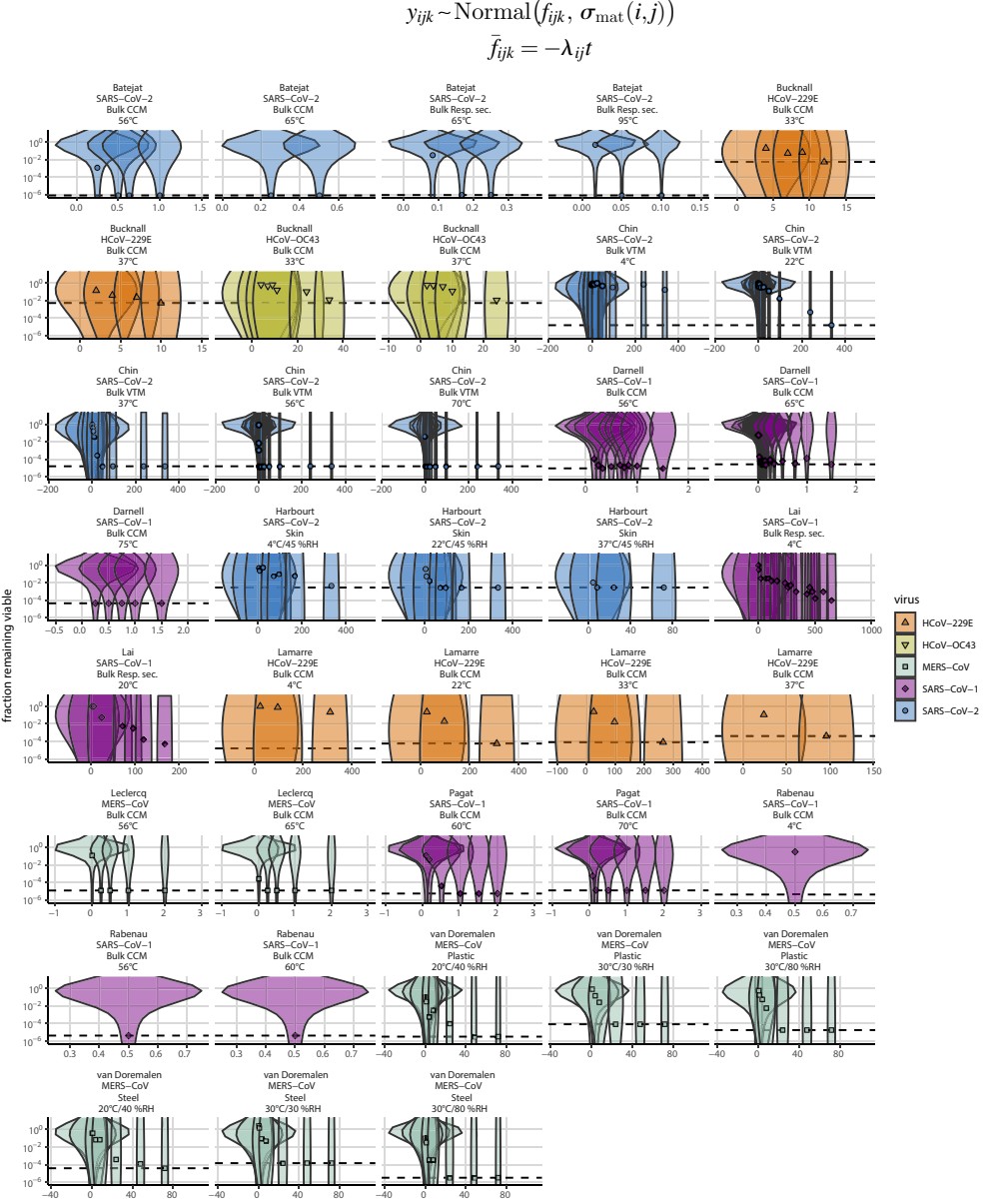

**Appendix 1—figure 21.** Prior predictive check for empirical coronavirus decay from literature data. Violin plots show distribution of simulated fractions of virus remaining viable sampled from the prior predictive distribution. Points show estimated fraction remaining viable for each collected sample based on data extracted from the literature. Shape and color indicates virus. x-axis shows time since first available measure. Study author, virus, and experimental conditions—material, temperature, and relative humidity (RH)—indicated at the top of each panel. Black dotted line shows LOD for each experiment. Wide coverage of violins relative to datapoints shows that priors are agnostic over the values of interest, and that the priors regard both fast and slow decay rates as possible.

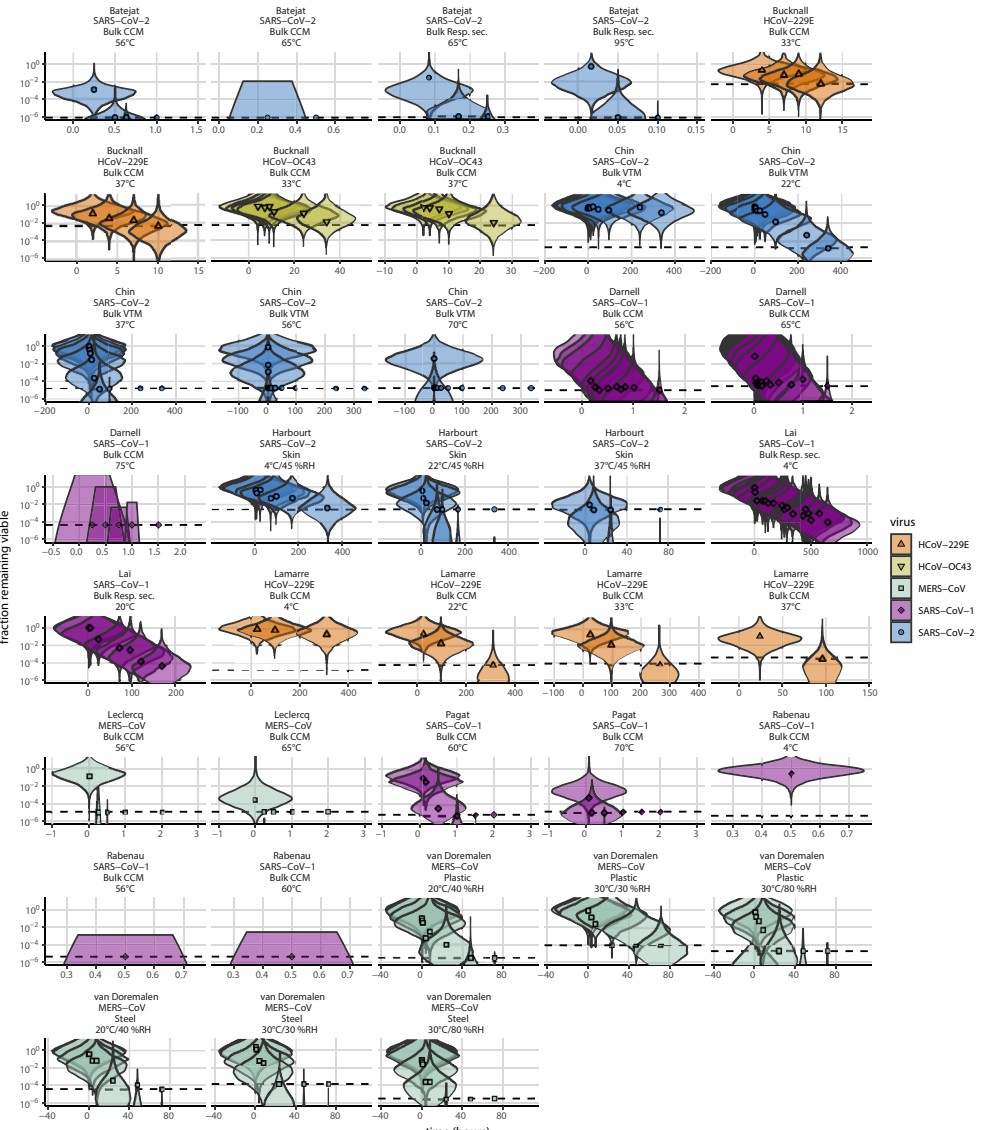

**Appendix 1—figure 22.** Posterior predictive check for empirical coronavirus decay from literature data. Violin plots show distribution of simulated fractions of virus remaining viable sampled from the posterior predictive distribution. Points show estimated fraction remaining viable for each collected sample based on data extracted from the literature. Shape and color indicates virus. x-axis shows time since first available measure. Study author, virus, and experimental conditions—material, temperature, and relative humidity (RH)—indicated at the top of each panel. Black dotted line shows LOD for each experiment. Tight correspondence between distribution of posterior simulated titers and independently estimated titers suggests the model fits the data well.

We placed a diffuse Normal prior on the log half-lives $\eta_{ij} = \frac{\log_{10}(2)}{\lambda_{ij}}$ and a Half-Normal prior on the standard deviations $\sigma_{\mathrm{mat}}(i,j)$:

$$\ln(\eta_{ij}) \sim \mathrm{Normal}(-2,4)$$
$$\sigma_{\mathrm{mat}}(i,j) \sim \mathrm{HalfNormal}(0.6,0.2) \tag{66}$$

## Estimation model predictive checks

We assessed appropriateness of priors with prior predictive checks (*Appendix 1—figure 21*) and goodness-of-fit with posterior predictive checks (*Appendix 1—figure 22*). Prior checks suggested

that prior distributions were agnostic over the parameter values of interest, and posterior checks suggested a good fit of the model to the data.

## Additional SARS-CoV-1 and MERS-CoV data

As noted in the Main Text Materials and methods, we made half-life estimates for SARS-CoV-1 and MERS-CoV at 22°C and 40% RH during the evaporation and quasi-equilibrium phases using data collected by our group during previous studies (*van Doremalen et al., 2020*). We included these estimates in the meta-analysis alongside the estimates described above. *Appendix 1—table 4* shows the estimated half-lives for these data, and *Appendix 1—figure 23* shows the fit of the simple regression model to these data.

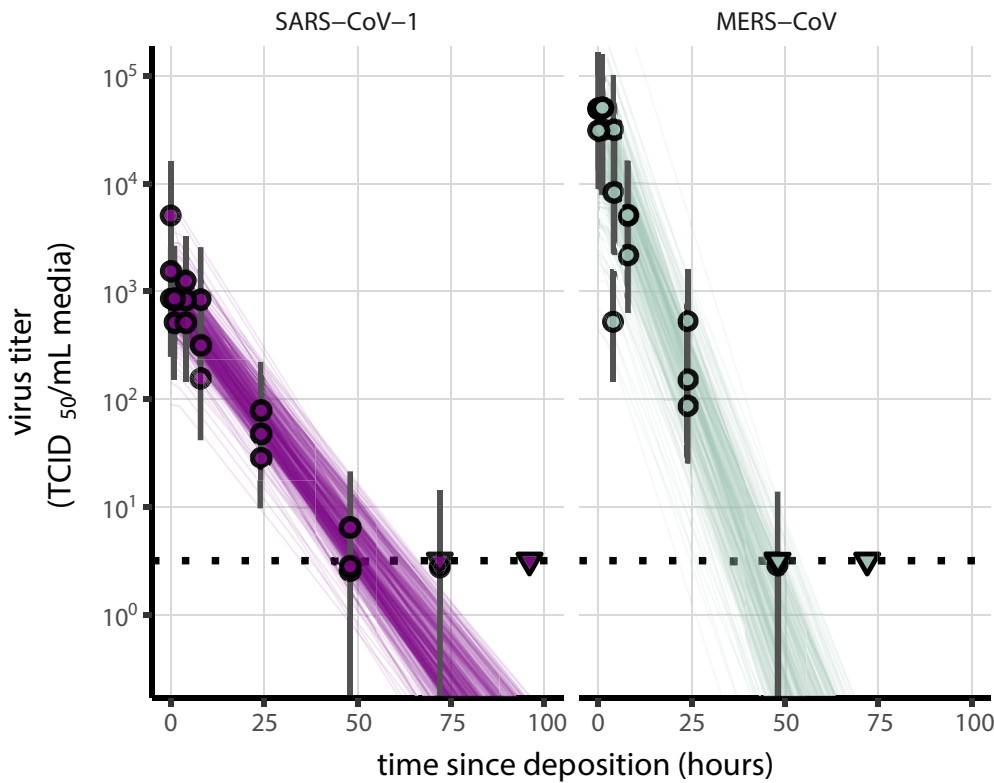

**Appendix 1—figure 23.** Fit of simple regression model to SARS-CoV-1 and MERS-CoV data. Points show posterior median estimated titers in $\log_{10}$ $TCID_{50}$/mL for each sample; lines show 95% credible intervals. Time-points with no positive wells for any replicate are plotted as triangles at the approximate single-replicate limit of detection (LOD) of the assay—denoted by a black dotted line at $10^{0.5}$ $TCID_{50}$/mL media—to indicate that a range of sub-LOD values are plausible. Three samples collected at each time-point. x-axis shows time since sample deposition. Lines are random draws (10 per sample) from the joint posterior distribution of the initial sample virus concentration and the estimated decay rate; the distribution of lines gives an estimate of the uncertainty in the decay rate and the variability of the initial titer for each experiment.

**Appendix 1—table 4.** Estimated half-lives in hours of SARS-CoV-1 and MERS-CoV on polypropylene as a function of temperature (T) and relative humidity (RH).
Estimated half-lives are reported as posterior median and the middle 95% credible interval.

|  | T (°C) | RH (%) | Virus | Median half-life (h) | 2.5 % | 97.5 % |
|---|---|---|---|---|---|---|
| Quasi-equilibrium phase | 22 | 40 | SARS-CoV-1 | 6.42 | 5.22 | 7.92 |
|  | 22 | 40 | MERS-CoV | 3.16 | 2.53 | 3.97 |
| Evaporation phase | 22 |  | SARS-CoV-1 | 11.55 | 1.43 | 207.68 |
|  | 22 |  | MERS-CoV | 13.18 | 1.09 | 217.34 |

## Meta-analysis estimates of half-lives

### Mechanistic model prediction of half-lives from literature

#### Absolute predictions

Where both temperature and humidity were available for a measurement from the literature, we were able to predict the absolute half-life directly from our main model fit, as parametrized from our own SARS-CoV-2 data. These predictions are plotted in Main Text *Figure 3c* and *Figure 3—figure supplement 1*.

#### Relative predictions

For many studies, however, only temperature information was available. Moreover, heterogeneities both among viruses and among laboratory protocols could shift half-lives by a constant factor relative to our SARS-CoV-2-polypropylene-DMEM data. To account for this, we made within-study relative predictions for studies with at least two temperature and/or humidity conditions on the same side of the ERH for a given virus on a given surface. For each such set of experiments, we chose the experiment whose temperature was closest to 20°C to serve as the reference experiment. If there were multiple such experiments, we picked the experiment with the relative humidity closest to the ERH.

Our mechanistic model implies that the ratio of a pair of half-lives $\eta_1$ and $\eta_2$ at ambient temperatures $T_1$ and $T_2$ and super-ERH relative humidities $h_1$ and $h_2$ is given by:

$$\frac{\eta_1}{\eta_2} = \left(\frac{\ln(h_2)}{\ln(h_1)}\right)^{\frac{1}{\alpha_c}} \exp\left[\frac{E_a}{R}\left(\frac{1}{T_1} - \frac{1}{T_2}\right)\right] \tag{67}$$

If $h_1$ and $h_2$ are both sub-ERH, we have:

$$\frac{\eta_1}{\eta_2} = \exp\left[\frac{E_a}{R}\left(\frac{1}{T_1} - \frac{1}{T_2}\right)\right] \tag{68}$$

Where no information about ambient relative humidity was available, we assumed humidities were shared across experiments and were super-ERH, and therefore used *Equation 67* with $h_1 = h_2$ to make predictions. Note that these predictions are independent of $\alpha_s$ and $A$; they rely only on relative rates of inactivation, not absolute ones. These relative predictions according to *Equation 67* and *Equation 68* are plotted in *Figure 3d*.

### Discussion of the results

We report half-life estimates for each experimental condition in *Appendix 1—table 3*. This meta-analysis highlights the same qualitative effect of temperature as our data: higher temperatures are associated with faster virus decay (shorter half-lives), with SARS-CoV-2 half-life in bulk medium varying from several hours at 4°C to less than 15 s at 95°C. The direct comparison of coronavirus half-lives across humidities is difficult, as only a few studies measured virus decay at several humidities with a fixed temperature.

This data set includes data collected following heterogeneous experimental procedures, which can considerably impact virus inactivation kinetics. For instance, we included data collected from suspensions at different pH, which notably explains the difference between the half-lives estimated from *Bucknall et al., 1972* (cell culture medium at pH 7.4) and *Lamarre and Talbot, 1989* (cell culture medium supplemented to reach pH 6) for HCoV-229E in bulk medium at 33°C and 37°C. Indeed, *Lamarre and Talbot, 1989* showed that pH 6 is optimal for HCoV-229E stability, hence the higher half-lives reported by this study. We also included data collected from suspensions supplemented with varying levels of proteins (from 1% [*Pagat et al., 2007*] to 10% [*Darnell et al., 2004*; *Harbourt et al., 2020*] of fetal calf serum) although protein concentration is known to impact virus inactivation kinetics (*Yang et al., 2012*; *Pastorino et al., 2020*). Containers used to expose samples to environmental conditions can also impact virus inactivation rate, but this information is rarely reported (*Gamble et al., 2021*). Notably, the two SARS-CoV-2 points in Main Text *Figure 3d* that

show shorter-than-predicted half-lives are from heated bulk medium in closed vials, where inactivation is known to be rapid (*Gamble et al., 2021*).

Despite this heterogeneity of the data collection process, and the high uncertainty of some half-life estimates, we find good qualitative agreement between model predictions and model-free estimates (see Main Text, *Figure 3*, and *Figure 3—figure supplement 1*).

## Methodological implications for experimental studies on virus stability

The mechanisms by which humidity impacts virus stability have methodological implications for future experimental studies. First, since solute concentration plays a critical role in the decay of viable virus, studies interested in virus viability should either include a measure of solute concentration over time (ideally via medium evaporation or precise measurements of sample mass through time), or focus on the quasi-equilibrium phase (during which solute concentration can be assumed to be constant). Second, since the evaporative kinetics and the resultant solute environments depend on the composition of the initial suspension medium, quantitative estimates of duration of virus viability based on experiments conducted in different media should be compared with caution. In our meta-analysis, we were able to make accurate relative predictions of data from multiple artificial medium formulations as well as from bodily fluids; this suggests that the underlying mechanisms are robust to variation in suspension medium, though absolute durations may vary. Third, given the non-linear relationship between virus half-life and relative humidity, studies interested in the effect of humidity on virus viability should include a wide range of conditions at constant temperature, including both sub- and super-ERH conditions.

Code for titer estimation and model fitting is freely available the online data and code repository, and could readily be adapted to the study of other viruses.

