## [Decision Letter]

**Acceptance summary:**

The study offered a provocative, highly technical, and courageously cross-disciplinary approach to understanding environmental survival in viruses, including SARS-CoV-2. We applaud the rigor and care in the work, and thank you for submitting it to *eLife*. There is little doubt that this will be a very worthy and meaningful contribution to the literature. Even further, this work may transcend the COVID-19 pandemic, and serve as grand example of how one can rigorously integrate fields like biophysics with epidemiology.

**Decision letter after peer review:**

Thank you for submitting your article "Mechanistic theory predicts the effects of temperature and humidity on inactivation of SARS-CoV-2 and other enveloped viruses" for consideration by *eLife*. Your article has been reviewed by 2 peer reviewers, one of whom is a member of our Board of Reviewing Editors, and the evaluation has been overseen by a Senior Editor.

Please address the changes as outlined by the reviewers below, as they will make for an improved manuscript. We encourage the authors to address them, and more broadly, make this article as clear as possible. Both reviewers provided some insight into this, and strongly encourage the authors to pay close attention to this.

*Reviewer #1:*

In this manuscript, Morris and colleagues offer a mechanistic model, featuring chemistry and physics details, that explain the effects of temperature and humidity.

I found this manuscript to be highly relevant for any virus, but given the context-the COVID-19 pandemic-I find the results to be timely, relevant, and fantastically well conducted. The interaction between the epidemiological knowledge and biophysical precision is clear, and the implications are also very clear. The manuscript is well-written and organized.

I believe that this manuscript should be published, would make a worthy contribution to great body of SARS-CoV-2 and extended COVID-19 literature.

The recommended changes have to do with subtle editorial points, and efforts to make this work as clear as possible.

1. Figure 1: I believe that the authors should check the labels. I could be wrong, but I believe the labels in the captions are wrong.

2. Equations:

The interaction between the text and the appendix could be improved, and I challenge the authors to better integrate this information. The authors should read the equations from the perspective of the reader: can the reader fully grasp the flow of information from the equations as provided. I am of a mixed opinion on this, but given the overall excellence of the manuscript, I encourage the authors to make this work as clear and digestible as possible, as it has the potential to be a defining piece of literature.

*Reviewer #2:*

This is a very well-done and thorough analysis of SARS-CoV-2 decay rates, with new theory and models being developed that also apply to other viruses. The methodology is solid, the analyses extensive, and the authors fully achieve their goal of further understanding the decay characteristics of SARS-CoV-2 and other viruses in the environment. The main novel contribution is the detailed mechanistic model relating decay rates to temperature and relative humidity. I believe this is a major contribution toward better understanding and predicting environmental virus decay dynamics.

I consider this paper perfectly suited for *eLife*. Overall, I think another pass-through the main manuscript with an eye to making it as easy and stand-alone to follow as possible for the 'casual reader/browser' would be useful.

---

## [Author Response]

Reviewer #1:In this manuscript, Morris and colleagues offer a mechanistic model, featuring chemistry and physics details, that explain the effects of temperature and humidity.I found this manuscript to be highly relevant for any virus, but given the context-the COVID-19 pandemic-I find the results to be timely, relevant, and fantastically well conducted. The interaction between the epidemiological knowledge and biophysical precision is clear, and the implications are also very clear. The manuscript is well-written and organized.I believe that this manuscript should be published, would make a worthy contribution to great body of SARS-CoV-2 and extended COVID-19 literature.The recommended changes have to do with subtle editorial points, and efforts to make this work as clear as possible.1. Figure 1: I believe that the authors should check the labels. I could be wrong, but I believe the labels in the captions are wrong.2. Equations:The interaction between the text and the appendix could be improved, and I challenge the authors to better integrate this information. The authors should read the equations from the perspective of the reader: can the reader fully grasp the flow of information from the equations as provided. I am of a mixed opinion on this, but given the overall excellence of the manuscript, I encourage the authors to make this work as clear and digestible as possible, as it has the potential to be a defining piece of literature.

We have revised the manuscript in light of the decision comments, with the aim of improving readability and clarity. Edits include collating supplementary figures as figure supplements to the main figures, where appropriate, nudging points in figures to avoid overplotting and improve readability, and clarifying distinctions between the different measurements and model fits used.